# Hydrogen production by *Sulfurospirillum* species enables syntrophic interactions of Epsilonproteobacteria

Stefan Kruse[1], Tobias Goris [1], Martin Westermann[2], Lorenz Adrian [3,4] & Gabriele Diekert[1]

Hydrogen-producing bacteria are of environmental importance, since hydrogen is a major electron donor for prokaryotes in anoxic ecosystems. Epsilonproteobacteria are currently considered to be hydrogen-oxidizing bacteria exclusively. Here, we report hydrogen production upon pyruvate fermentation for free-living Epsilonproteobacteria, *Sulfurospirillum* spp. The amount of hydrogen produced is different in two subgroups of *Sulfurospirillum* spp., represented by *S. cavolei* and *S. multivorans*. The former produces more hydrogen and excretes acetate as sole organic acid, while the latter additionally produces lactate and succinate. Hydrogen production can be assigned by differential proteomics to a hydrogenase (similar to hydrogenase 4 from *E. coli*) that is more abundant during fermentation. A syntrophic interaction is established between *Sulfurospirillum multivorans* and *Methanococcus voltae* when cocultured with lactate as sole substrate, as the former cannot grow fermentatively on lactate alone and the latter relies on hydrogen for growth. This might hint to a yet unrecognized role of Epsilonproteobacteria as hydrogen producers in anoxic microbial communities.

[1] Department of Applied and Ecological Microbiology, Institute of Microbiology, Friedrich Schiller University, Philosophenweg 12, 07743 Jena, Germany. [2] Center for Electron Microscopy of the University Hospital Jena, Ziegelmühlenweg 1, 07743 Jena, Germany. [3] Department Isotope Biogeochemistry, Helmholtz Centre for Environmental Research—UFZ, Permoserstr. 15, 04318 Leipzig, Germany. [4] Fachgebiet Geobiotechnologie, Technische Universität Berlin, Ackerstraße 76, 13355 Berlin, Germany. These authors contributed equally: Stefan Kruse, Tobias Goris. Correspondence and requests for materials should be addressed to T.G. (email: tobiasgoris@gmail.com)

Hydrogen gas ($H_2$), an important energy substrate for many bacteria and archaea, plays a crucial role in the anaerobic food web, e.g. in syntrophic interactions. It is produced by fermenting bacteria as a result of the disposal of excess reducing equivalents. Besides $H_2$, also formate, similarly formed during fermentative metabolism, is an important electron carrier in e.g. syntrophic fatty acid-degrading methanogenic consortia[1]. Other prokaryotes may use both $H_2$ and formate as an electron donor for e.g. sulfate respiration or methanogenesis. In syntrophic interactions, the formate-/$H_2$-producing bacterium is dependent on the electron donor uptake by its syntrophic partner, which sustains a low $H_2$ partial pressure or low formate concentration and thus enables $H_2$/formate production, which would otherwise thermodynamically be unfavorable[2–4]. For example, butyrate, propionate or acetate-oxidizing anaerobic bacteria that form $H_2$ or formate as fermentation product are dependent on formate-/$H_2$-oxidizing microorganisms such as methanogenic archaea[5–7]. It was shown that the interspecies $H_2$ or formate transfer becomes more efficient when syntrophs and methanogens are in close physical contact[8,9]. The syntrophic degradation of propionate by a coculture of *Pelotomaculum thermopropionicum* and *Methanothermobacter thermoautotrophicus* as well as butyrate degradation coupled to organohalide respiration by *Syntrophomonas wolfei* and *Dehalococcoides mccartyi* 195 resulted in aggregate formation and cell-to-cell contact of the involved organisms[10,11]. Besides interspecies transfer of molecular energy carriers, electrons can be transferred directly between syntrophic partners via electrodonductive protein connections in a process termed direct interspecies electron transfer[12]. In addition to the importance of $H_2$ in microbial food webs, $H_2$ is considered to be an alternative energy source and biohydrogen production by microorganisms is discussed as one way to generate environmentally compatible fuels[13].

Epsilonproteobacteria are hitherto considered to be $H_2$-consuming organisms and $H_2$-oxidizing enzymes of only a few Epsilonproteobacteria are characterized so far, e.g. the membrane-bound uptake hydrogenases of *Helicobacter pylori* and *Wolinella succinogenes*[14,15]. *Sulfurospirillum carboxydovorans* was shown to produce minor amounts of hydrogen, which was finally consumed again, upon CO oxidation[16]. Fermentative $H_2$ production has never been shown to be performed by any Epsilonproteobacterium so far, although in recent years several Epsilonproteobacteria, especially marine, deep vent-inhabiting species, were reported to encode putative $H_2$-evolving hydrogenases in their genomes[17–25]. *Sulfurospirillum* spp. are free-living, metabolically versatile Epsilonproteobacteria, many of which are known for their ability to respire toxic or environmentally harmful compounds such as arsenate, selenate or organohalides (e.g. tetrachloroethene—PCE)[26,27]. The anaerobic respiration with PCE, leading to the formation of *cis*-1,2-dichloroethene (cDCE), was studied in detail in *Sulfurospirillum multivorans* (formerly known as *Dehalospirillum multivorans*)[28,29]. Several *Sulfurospirillum* spp. were found in contaminated sediments, wastewater plants, marine environments or on biocathodes[16,22,26,30]. The role of *Sulfurospirillum* in such environments is unclear.

In previous studies, four gene clusters, each encoding a [NiFe] hydrogenase, were found in the genome of *S. multivorans*[23] and most other *Sulfurospirillum* spp.[26]. Two of these appear to be $H_2$-producing, the other two are potential $H_2$-uptake enzymes as deduced from sequence similarity to known hydrogenases. Of these four hydrogenases, one of each type, $H_2$-oxidizing and $H_2$-producing, were previously detected in *S. multivorans*[29,31]. The periplasmically oriented $H_2$-oxidizing enzyme is very similar to the characterized *W. succinogenes* and *H. pylori* membrane-bound hydrogenases (MBH). It comprises three subunits, the large subunit, harboring the NiFe active site, a small subunit for electron transfer with three FeS clusters, and a membrane-integral cytochrome *b*. The putative $H_2$-producing, cytoplasmically oriented enzyme (Hyf) is a large, complex enzyme with eight subunits, four of them presumably membrane-integral. Regarding amino acid sequence and subunit architecture, this hydrogenase is similar to hydrogenase 4 of *Escherichia coli*, part of a putative second formate hydrogen lyase (FHL)[32]. However, in *S. multivorans*, Hyf is unlikely to form an FHL complex since the corresponding gene cluster does not encode any formate-specific proteins as is the case for the FHL complexes in *E. coli* (Supplementary Figure 1).

Here, we show that several *Sulfurospirillum* spp. produce $H_2$ upon pyruvate fermentation. *Sulfurospirillum cavolei* was observed to produce more $H_2$ than other *Sulfurospirillum* spp., which is caused by a different fermentation metabolism. To unravel the metabolism and the hydrogenase equipment of both organisms, label-free comparative proteomics was carried out. A coculture experiment of *S. multivorans* with the methanogenic archaeon *Methanococcus voltae* revealed an interspecies $H_2$ transfer between both organisms suggesting a hitherto undiscovered contribution of *Sulfurospirillum* spp. and other Epsilonproteobacteria to the microbial anaerobic food web as $H_2$ producers.

## Results

**Adaptation of *S. multivorans* to pyruvate fermentation**. In previous studies, *S. multivorans* and other *Sulfurospirillum* spp. were shown to grow fermentatively on pyruvate[26,33,34]. Only few data on growth behavior are available in the literature, but *S. multivorans* was reported to exhibit poor growth on pyruvate as sole energy source compared to respiratory growth with pyruvate and fumarate or tetrachloroethene (PCE) as electron acceptor[33]. However, we observed an adaptation of *S. multivorans* to fermentative growth on pyruvate. After about 20 transfers with 10% inoculum each, a growth rate of 0.09 h$^{-1}$ was determined (growth rate on pyruvate/fumarate, 0.19 h$^{-1}$, Fig. 1, Supplementary Figure 2). During the adaptation to pyruvate fermentation, the growth rate increased on average by 0.02 h$^{-1}$ with each transfer (Supplementary Figure 3). In addition, the lag phase duration decreased from initially 40 to 5 h. After 18 transfers, no further significant increase of the growth rate was observed. This adaptation process was also observed for *S. cavolei*, *Sulfurospirillum deleyianum* and *Sulfurospirillum arsenophilum*. For further investigation of the adaptation process, *S. multivorans* cells not adapted to pyruvate fermentation and those adapted to pyruvate fermentation were harvested and subjected to proteome analysis (see subsection "Comparative genomics and proteomics" under the Results section).

**Fermentative growth and $H_2$ production by *Sulfurospirillum***. To get deeper insight into the fermentation pathways and $H_2$ production capabilities of *Sulfurospirillum* spp., several species were cultivated with pyruvate as sole substrate. Six species were tested for pyruvate fermentation, of which *Sulfurospirillum barnesii* and *Sulfurospirillum halorespirans* were not able to grow even after cultivation for several months. *S. cavolei*, *S. deleyianum*, and *S. arsenophilum* grew on pyruvate alone at slower rates than *S. multivorans* (0.03, 0.06, and 0.004 h$^{-1}$, respectively, Fig. 2, Supplementary Figures 4 and 5). $H_2$ production was measured for all fermentatively growing *Sulfurospirillum* spp., but the produced amount differed, depending on the species. *S. cavolei* produced the highest amount of $H_2$ followed by *S. arsenophilum*. *S. deleyianum*, and *S. multivorans* produced about 100 μmol $H_2$ per 100 mL culture. *Desulfitobacterium hafniense* DCB-2, a known

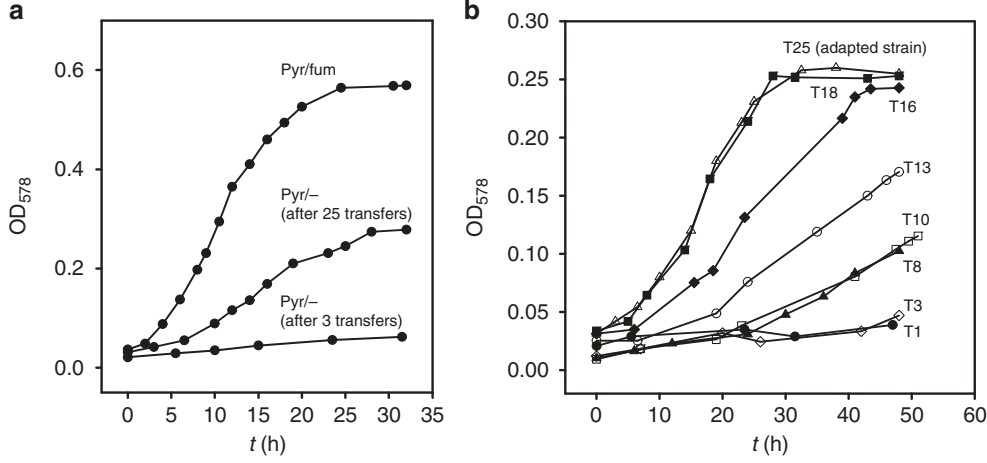

**Fig. 1** Adaptation of *S. multivorans* to pyruvate-fermenting conditions. **a** Growth curves with pyruvate as sole growth substrate after three and 25 transfers; a culture with pyruvate/fumarate after three transfers is shown for comparison. **b** Growth during continuous transfer on pyruvate without electron acceptor. Each transfer (10% inoculum) was done after 48 h cultivation. Data were obtained from at least two independent biological replicates and are representatives; graphs of the replicates are shown in Supplementary Figure 2. T number of transfer step, Pyr pyruvate, Fum fumarate, OD$_{578}$ optical density at 578 nm

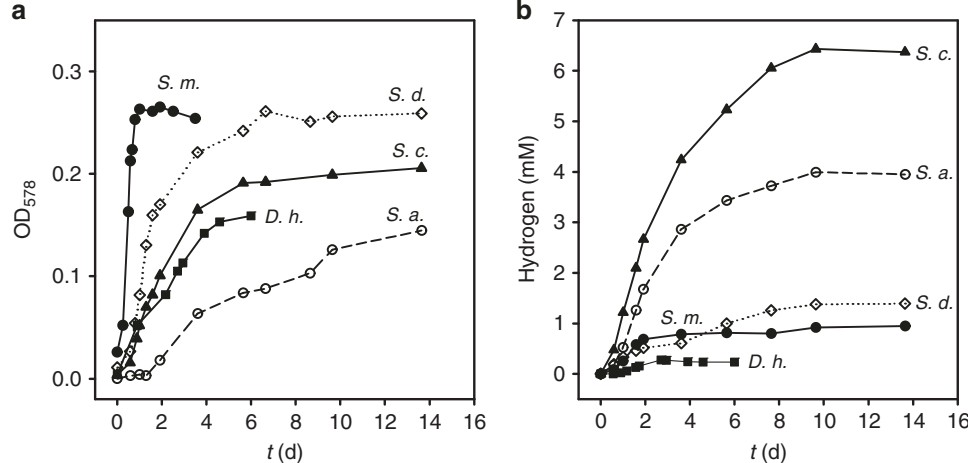

**Fig. 2** Growth and H$_2$ production during fermentative growth. The graphs show growth curves (**a**) and H$_2$ concentration (**b**) during pyruvate fermentation (40 mM pyruvate) by *Sulfurospirillum* spp. and *D. hafniense* strain DCB-2, each after adaptation. The graphs are representatives of three independent replicates; the graphs of the latter are presented in Supplementary Figures 4 and 5. S.m. *S. multivorans*, S.d. *S. deleyianum*, S.c. *S. cavolei*, S.a. *S. arsenophilum*, D.h. *D. hafniense* DCB-2

pyruvate-fermenting organohalide-respiring bacterium, grows similar to *Sulfurospirillum* spp. (Fig. 2a) but produced only minor amounts of H$_2$ (20 μmol) (Fig. 2b). Fermentative growth on lactate was not observed for any of the organisms including *D. hafniense* DCB-2 even after cultivation for several months.

**Fermentative metabolism of *S. multivorans* and *S. cavolei*.** To unravel the fermentative metabolism of two *Sulfurospirillum* spp. showing different H$_2$ production patterns during growth on pyruvate, *S. multivorans* and *S. cavolei* were cultivated in a fermentation apparatus in which the gas phase of the Schott bottle was connected to CO$_2$ and H$_2$ traps (see Supplementary Figure 6) to avoid increasing gas partial pressures and hence a possible product inhibition on H$_2$ production or growth (see also subsection "Product inhibition by H$_2$ on fermentation of *Sulfurospirillum*"). Enhanced H$_2$ evolution was measured when compared to the serum bottle experiment, with up to hundred times more H$_2$ produced, while the growth was slower than in the previous setup (Fig. 3a, Supplementary Figure 7A and 8A). After consumption of 40 mM pyruvate, 27 mM acetate, 10 mM lactate,

3 mM succinate, 10 mM H$_2$, and 28 mM CO$_2$ were measured as fermentation products of *S. multivorans* (Fig. 3a). *S. cavolei* showed slower growth than *S. multivorans* and a much higher amount of H$_2$ evolved. During growth, which took 8−10 days, pyruvate (40 mM) was used up completely and 38 mM acetate, 36 mM H$_2$, and 38 mM CO$_2$ were the only products detected (Fig. 3b, Supplementary Figures 7B and 8B). No other organic acids such as formate or alcohols, e.g. ethanol, were detected for both. *S. deleyianum* showed similar fermentation products as observed with *S. multivorans* (Supplementary Figure 9).

The stoichiometry of the fermentation was verified by calculating the carbon recovery and an oxidation/reduction balance (Supplementary Table 1, Eqs. (1) and (2)). In *S. multivorans*, the amount of reducing equivalents generated from pyruvate oxidation was calculated to be 54 [H], which fits to the amount of used reducing equivalents for the production of molecular hydrogen, lactate and succinate (52 [H], Supplementary Table 1). In *S. cavolei*, pyruvate oxidation leads to the generation of 76 [H], which were almost exclusively (72 [H]) used for proton reduction to H$_2$. In addition, the carbon recovery is in

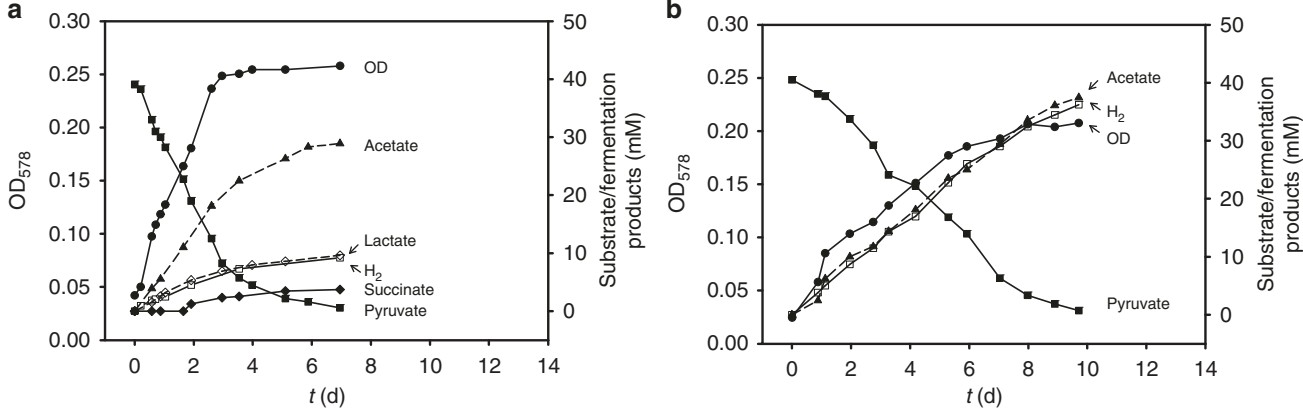

**Fig. 3** Fermentation balances of *Sulfurospirillum* spp. during fermentation. Substrate and fermentation product concentrations of *S. multivorans* (**a**) and *S. cavolei* (**b**) during fermentative growth on 40 mM pyruvate as measured in the fermentation apparatus. Graphs are representative of three independent biological replicates; the replicate graphs are shown in Supplementary Figures 7 and 8

agreement with the theoretical values and is 102.5% for *S. multivorans* and 95% for *S. cavolei*. The anabolic assimilation of the carbon source is minor with approximately 2.5 mM for *S. multivorans* and 2 mM for *S. cavolei* as calculated from OD and dry weight.

$$1.0 \, \text{Pyruvate} \rightarrow 0.7 \, \text{Acetate} + 0.25 \, \text{Lactate} + 0.075 \, \text{Succinate}$$
$$+ 0.25 \, \text{H}_2 + 0.7 \, \text{CO}_2 \tag{1}$$

$$1.0 \, \text{Pyruvate} \rightarrow 0.95 \, \text{Acetate} + 0.9 \, \text{H}_2 + 0.95 \, \text{CO}_2 \tag{2}$$

**Product inhibition by H₂ on fermentation of *Sulfurospirillum*.** The different amount of H₂ produced in the growth experiments in serum bottles and the fermentation apparatus imply a product inhibition of H₂ on H₂ production. To investigate the effect of H₂ in the gas phase on the fermentative growth of *S. multivorans* and *S. cavolei*, both organisms were cultivated in serum bottles with a gas phase of 100% H₂ or 100% nitrogen (Fig. 4, Supplementary Figures 10 and 11). With nitrogen as gas phase, *S. multivorans* and *S. cavolei* showed similar growth and production rates of organic acids as observed in the fermentation apparatus. A strong negative effect on growth was observed with 100% H₂ in the gas phase. *S. multivorans* was still able to ferment pyruvate but showed an inhibited growth and a lower cell density compared to the culture without H₂ in gas phase, while *S. cavolei* was almost completely inhibited (Fig. 4a). The restricted growth is also reflected by a lower pyruvate consumption rate (Fig. 4b). In addition, the formation of fermentation products shifted from acetate production to lactate and succinate formation in *S. multivorans* (Fig. 4c–e). *S. cavolei* produced neither lactate nor succinate and only minor amounts of acetate.

**Hydrogenase activities by *Sulfurospirillum* cell suspensions.** The H₂ production and oxidation capability of cell suspensions of *S. multivorans* and *S. cavolei* was analyzed to obtain further evidence about the hydrogenase involved in the production and oxidation reaction. Transcriptional and proteomic studies revealed the presence of two [NiFe] hydrogenases in *S. multivorans*[31]: a hydrogen-oxidizing periplasmically oriented MBH and a putative H₂-producing cytoplasmically oriented MBH (Hyf). These two hydrogenases might be distinguished by their different subcellular localization and thus their accessibility to redox mediators like viologens in hydrogenase activity assays.

Photometrically measured H₂-oxidizing activity was detected in whole cell suspensions as well as in membrane and soluble fractions (Table 1). In contrast, H₂-producing activity, as monitored by GC, was only measured with membrane fractions but not in whole cell suspensions of *S. multivorans* and *S. cavolei* with approximately 1.5-fold higher activity in *S. cavolei* (Table 1). The membrane fractions of *S. multivorans* and *S. cavolei* cells grown on pyruvate as sole energy source were about 2-fold more active in H₂-production than those of cells cultivated under respiratory growth conditions with pyruvate plus fumarate, while the latter exhibited slightly more H₂ oxidation activity. *Clostridium pasteurianum* DSM 525, which is known to harbor a cytoplasmic soluble H₂-producing hydrogenase, exhibited hydrogenase activity only in the soluble fractions and showed no H₂ producing activity in cell suspensions with methyl viologen as electron donor (Supplementary Table 2), thus serving as a control for the hydrogenase localization experiment.

**Comparative genomics and proteomics.** To unravel the cause of the different fermentative metabolisms of the two *Sulfurospirillum* sp., a comparative genomic analysis was done with the RAST sequence comparison tool[35]. Additionally, proteomes of *S. cavolei* NBRC109482 and *S. multivorans* cultivated under fermenting and respiring conditions with fumarate as electron acceptor were analyzed. Bidirectional blast hits with more than 50% amino acid sequence identity were considered as orthologs, proteins putatively fulfilling the same functions in both organisms. The genomes were overall similar, with 2057 of 2768 of the encoded proteins in *S. cavolei* being orthologs. Only few of the nonorthologous proteins in *S. cavolei* could be considered to play a role in the fermentation based on their annotation and putative involvement in one of the pathways connected to fermentative catabolism. Among the proteins encoded in the *S. cavolei* genome (annotated RefSeq WGS accession number NZ_BBQE01000001.1), which do not have an ortholog in *S. multivorans*, we found a cluster encoding an [FeFe] hydrogenase (Supplementary Figure 12) known to contribute to fermentative H₂ production in many bacteria, e.g. Clostridia. A nearly identical gene cluster is found in the other two genomes of *S. cavolei* strains UCH003 (NZ_AP014724.1) and MES (JSEC00000000.1), the latter of which was assembled from a metagenome[22]. The large [FeFe] hydrogenase catalytic subunit gene, *hydA*, is disrupted by a stop codon resulting from a nucleotide insertion only in *S. cavolei*. The mutation was confirmed by PCR and Sanger sequencing. RT PCR analysis suggested that *hydA* was transcribed under pyruvate-fermenting growth conditions (Supplementary

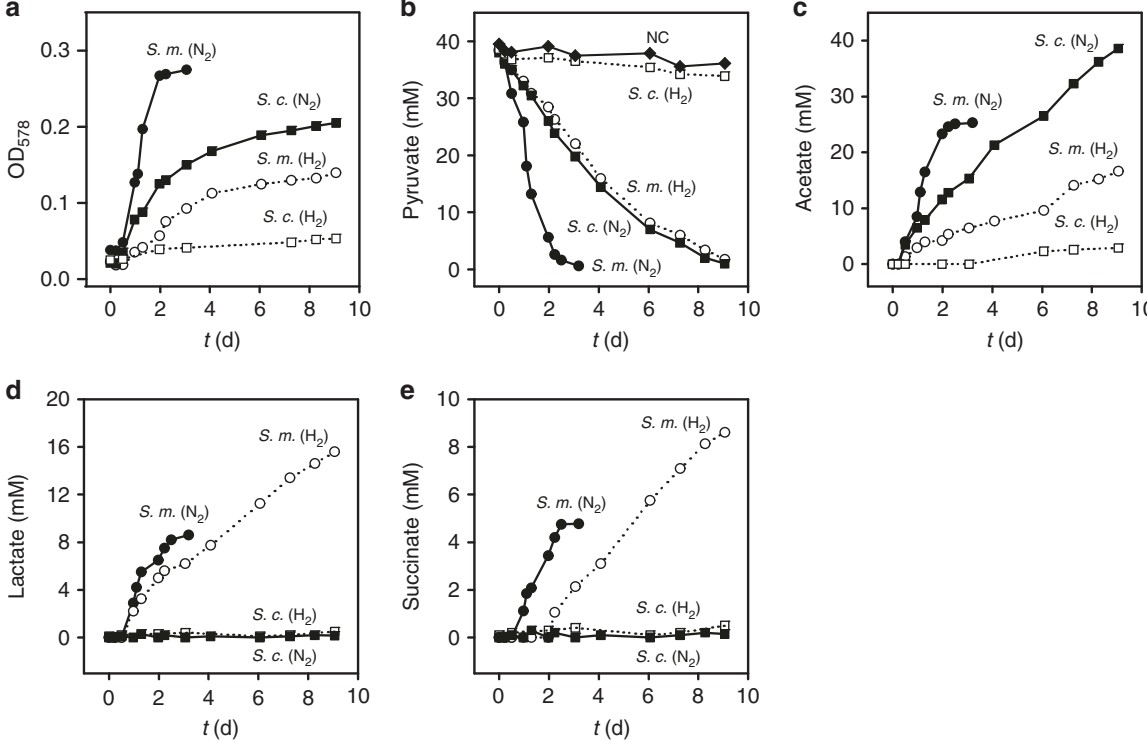

**Fig. 4** Growth and formation of fermentation products during cultivation under 100% nitrogen ($N_2$) and 100% $H_2$ atmosphere. Growth curve (**a**), pyruvate consumption (**b**) and acetate (**c**), lactate (**d**), and succinate (**e**) production during pyruvate fermentation are shown. Organic acids were measured via HPLC. Each cultivation was conducted in three biological replicates, shown in Supplementary Figures 10 and 11. S.m. *S. multivorans*, S.c. *S. cavolei*, $N_2$ nitrogen, $H_2$ hydrogen, NC negative control (cell-free medium)

**Table 1 Hydrogen-producing and oxidizing activities of cell suspensions and cellular fractions of *S. multivorans* and *S. cavolei* cultivated with pyruvate or with pyruvate/fumarate**

| Cellular fraction | Hydrogenase activity (nkat mg⁻¹) | | | | | |
|---|---|---|---|---|---|---|
| | ***S. multivorans*** | | | ***S. cavolei*** | | |
| | **MV → $H_2$** | **$H_2$ → BV** | **$H_2$ → MV** | **MV → $H_2$** | **$H_2$ → BV** | **$H_2$ → MV** |
| Cell suspensions | | | | | | |
| Pyr | <0.01 | 4.1 ± 0.5 | 0.7 ± 0.3 | <0.01 | 5.5 ± 0.6 | 1.4 ± 0.3 |
| Membrane fractions | | | | | | |
| Pyr | 12.3 ± 2.4 | 23.5 ± 2.1 | n.d. | 20.6 ± 3.7 | 10.1 ± 0.5 | n.d. |
| Pyr + Fum | 5.7 ± 1.5 | 36.6 ± 3.3 | n.d. | 10.1 ± 2.4 | 13.5 ± 1.6 | n.d. |
| Soluble fractions | | | | | | |
| Pyr | <0.01 | n.d. | n.d. | <0.01 | n.d. | n.d. |
| Pyr + Fum | <0.01 | 2.3 ± 0.3 | n.d. | <0.01 | 1.9 ± 0.3 | n.d. |

MV → $H_2$ indicates $H_2$ formation activity, $H_2$ → BV/MV indicates $H_2$ oxidation
Data are derived from three independent biological replicates and show means ± standard deviation
*MV* methyl viologen, *BV* benzyl viologen, *Pyr* pyruvate, *Fum* fumarate, *n.d.* not determined

Figure 13). However, the [FeFe] hydrogenase was not identified in the proteome of *S. cavolei*.

Apart from the [FeFe] hydrogenase, the four [NiFe] hydrogenase gene clusters were highly similar in the genomes of both *S. multivorans* and *S. cavolei*. The Hyf hydrogenase was found in high abundances especially in the proteome of *S. multivorans* cultivated with pyruvate alone. Here, four out of eight of the structural subunits were found in the 10% of the most abundant proteins, while none were found in the top 10% under respiratory conditions. In *S. cavolei*, the hydrogenase 4 subunits were not as abundant as in *S. multivorans* with only two out of six quantified subunits in the top 20% (Supplementary Data 1 and 2). In both

organisms, a significantly higher amount of Hyf subunits was quantified under fermentative growth conditions (*S. multivorans*: 4- to 27-fold for the structural subunits HyfA-HyfI, all *p* values are <0.001, *S. cavolei*: 2- to 5-fold for HyfA-HyfI, all *p* values are <0.05; Fig. 5, Supplementary Table 3, Supplementary Data 2). Interestingly, the *Hyf* gene cluster is disrupted at one site in *S. halorespirans*, which cannot grow on pyruvate alone. Genome sequencing[36] (CP017111.1) revealed a transposase insertion at *hyfB* which might result in a nonfunctional gene *S. halorespirans*. The transposon insertion was confirmed by PCR using *hyfB*-specific primers flanking the transposase (Supplementary Figure 14, Supplementary Table 4). The membrane-bound subunits

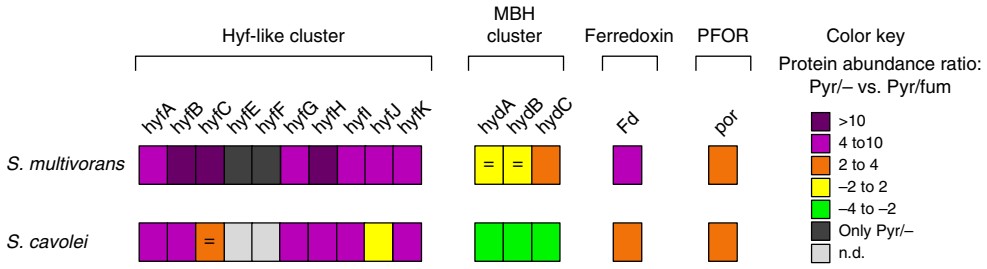

**Fig. 5** Comparative proteomics of proteins possibly involved in pyruvate fermentation of *S. multivorans* and *S. cavolei*. Comparison of cells grown with pyruvate alone was done with cells grown with pyruvate/fumarate. For quantified proteins the protein intensity ratio is given as colored squares. Nonsignificantly altered proteins levels are marked with an equal sign (*p* values > 0.05). Proteins exclusively found in pyruvate-fermenting cells are colored dark gray. All data were obtained from three independent biological replicates. Hyf-like Hyf hydrogenase (SMUL_2383–2392; SCA02S_RS01920-RS01965), MBH membrane-bound hydrogenase (SMUL_1423–1425; SCA02S_RS01350-RS01360), Fd ferredoxin (SMUL_0303; SCA025_RS12260), PFOR pyruvate:ferredoxin oxidoreductase (SMUL_2630; SCA02S_RS04525), Pyr pyruvate, Fum fumarate

HyfE and HyfF were found in fermenting cells of *S. multivorans* exclusively. Sequence comparison of the Hyf hydrogenase of *S. multivorans* shows similarities to the proton-pumping complex I of *Thermus thermophilus* (Supplementary Figure 15). An analysis regarding the potential proton-pumping capabilities of the *S. multivorans* Hyf deduced from conserved amino acids which are responsible for proton pumping in complex I of *T. thermophilus* and a comparison to the *E. coli* FHL is given in the Supplementary information (Supplementary Note 1, Supplementary Table 5 and Supplementary Figures 16–18). Important conserved amino acids that are likely involved in proton pumping of complex I are fully conserved in one of the four Hyf membrane subunits, namely in HyfF.

A search for the *hyf* gene cluster in genomes of Epsilonproteobacteria shows that it is ubiquitous in, but not limited to, *Sulfurospirillum* spp. (Supplementary Table 6). Four out of 15 *Sulfurospirillum* sp. genomes harbor a second *hyf* gene cluster colocated with genes encoding a formate transporter and a formate dehydrogenase (Supplementary Figure 1). In *Arcobacter* spp. and the marine species *Caminibacter mediatlanticus* and *Lebetimonas* spp., only the latter gene cluster encoding a putative FHL complex is found. In several *Campylobacter* spp. including *Campylobacter concisus*, a *hyf* gene cluster with a formate transporter gene was identified (Supplementary Figure 1), while a second group of *Campylobacter* (including *Campylobacter fetus*) does not encode any formate-related proteins (Supplementary Table 6).

The subunits of the MBH were quantified in either unsignificantly lower amounts (HydAB, approximately 2-fold, *p* values 0.40 and 0.07, Fig. 5) or slightly higher amounts (HydC, approximately 2-fold, *p* value 0.01) under fermenting conditions for *S. multivorans*. In contrast, HydABC were found in significantly lower amounts in *S. cavolei* when grown fermentatively (Fig. 5). Of the cytoplasmic $H_2$-producing hydrogenase (Ech-like), only one subunit (present in the lower 50% abundant proteins) was quantified in *S. multivorans* grown with pyruvate alone. In *S. cavolei*, five of six Ech-like hydrogenase subunits were quantified in cells cultivated with pyruvate alone and two of six subunits in pyruvate/fumarate-cultivated cells, all of them in the lowest third abundant proteins. No subunit of the cytoplasmic uptake hydrogenase (HupSL) was found in any of the proteomes.

Of the proteins related to pyruvate metabolism, a pyruvate, water dikinase (phosphoenolpyruvate [PEP] synthetase) is encoded in the genome of *S. multivorans* (encoded by SMUL_1602), but not in *S. cavolei*. This enzyme is responsible for the ATP-dependent synthesis of phosphoenolpyruvate from pyruvate in gluconeogenesis (Supplementary Figure 19). The PEP synthetase was found in 6.3-fold higher abundance (*p* value 0.02)

in the proteome of fermentatively cultivated *S. multivorans* cells (Supplementary Table 3). In *S. cavolei*, PEP might be formed from pyruvate via oxaloacetate by two reactions catalyzed by pyruvate carboxylase and PEP carboxykinase. These two enzymes are encoded in one gene cluster (SCA02S_RS02520 and SCA02S_RS02525, respectively, Supplementary Figure 20). In *S. multivorans* these proteins (SMUL_0789 and SMUL_0791) cluster with a gene encoding a subunit similar to the membrane subunit of a putative $Na^+$-translocating oxaloacetate decarboxylase (SMUL_0790), of which an ortholog is not encoded in *S. cavolei* (Supplementary Figure 20). Both pyruvate carboxylase/oxaloacetate decarboxylase and PEP carboxykinase were found in the proteomes of both organisms in slightly higher amounts in cells grown with pyruvate only (Supplementary Data 2). Similar to *S. cavolei*, also *S. arsenophilum*, producing larger amounts of $H_2$ than *S. multivorans* (Fig. 2), lacks the putative oxaloacetate decarboxylase subunit gene.

A pyruvate:ferredoxin oxidoreductase (PFOR) and a ferredoxin (Fd) showed also a higher abundance in both *Sulfurospirillum* sp. under fermenting conditions (*S. multivorans*: PFOR 2-fold, Fd 6-fold, *S. cavolei*: PFOR 4-fold, Fd 2-fold, all *p* values are <0.01; Fig. 5, Supplementary Table 3). A second pyruvate-oxidizing enzyme, a quinone-dependent pyruvate dehydrogenase encoded exclusively in the genome of *S. multivorans*, was significantly lower abundant during pyruvate fermentation (7-fold, *p* value 0.02). The enzymes responsible for ATP generation via substrate-level phosphorylation, phosphotransacetylase and acetate kinase are slightly higher abundant during pyruvate fermentation in both *Sulfurospirillum* sp. (approximately 2-fold for both enzymes in *S. multivorans*, *p* values are <0.01 and approximately 3-fold in *S. cavolei*, *p* values are <0.001; Supplementary Table 3). The malic enzyme is higher abundant during fermentation in *S. multivorans* (3.7-fold, *p* value <0.001, Supplementary Table 3) and not quantified in any proteome of *S. cavolei* (Supplementary Table 3).

A putative lactate dehydrogenase (SMUL_0438, SCA02S_RS08360) with 35% amino acid sequence identity to a characterized lactate-producing lactate dehydrogenase from *Selenomonas ruminantium*[37] was not detected in these proteomes. This is in accordance to the lack of pyridine dinucleotide-dependent lactate-oxidizing or pyruvate-reducing activities in cell extracts of *S. multivorans* (methods described in Supplementary Note 2). Several candidates for pyridine dinucleotide-independent lactate dehydrogenases (iLDH) are encoded in the genome of *S. multivorans*. Since *S. deleyianum* shows also lactate production during pyruvate fermentation, only genes present as orthologs in both genomes were considered to be responsible for lactate production in *Sulfurospirillum* spp. Functionally characterized iLDHs are flavin and FeS-cluster-containing oxidoreductases[38] or

**Table 2 Significantly altered protein abundances in the proteomes of pyruvate-adapted cells compared to that of nonadapted cells**

| Sm Locus | Protein | Lg Pyr | Lg Pyr Ad | Ratio Pyr Ad/Pyr | Ratio Sm Ferm | Ratio Sc Ferm |
|---|---|---|---|---|---|---|
| SMUL_0150 | cytochrome *c* | 6.4 | 8.0 | 39.5 | 8.1 | 7.1 |
| SMUL_2101 | Aldehyde oxidoreductase | 7.7 | 9.0 | 16.5 | 141.6 | 6.6 |
| SMUL_2819 | Asparaginase | 8.5 | 9.3 | 6.1 | 4.1 | 3.8 |
| SMUL_3232 | NosL family protein | 7.2 | 8.7 | 31.4 | 5.9 | 6.1 |

Shown are the proteins which at the same time are significantly altered in the *S. multivorans* and *S. cavolei* proteomes of fermentatively cultivated cells
*Sm Sulfurospirillum multivorans*, *Sc Sulfurospirillum cavolei*, *Lg* Log10 value, *Pyr* pyruvate, *Ferm* fermentative metabolism vs. respiratory metabolism

enzymes related to malate:quinone oxidoreductase[39]. Only two candidates of the former class were identified in the genome, encoded by SMUL_1449 and SMUL_2229. Of these, only the latter gene product was detected in the proteome, however, not in altered amounts under fermentative conditions when compared to respiratory cultivation. To detect proteins involved in lactate oxidation, we compared the proteomes of *S. multivorans* cells cultivated with lactate and fumarate to that of pyruvate/fumarate-cultivated cells. While we did not observe significant changes in protein abundances of any proteins putatively involved in lactate oxidation, we observed a high abundance of a membrane-bound flavin and Fe-S cluster-containing protein (encoded by SMUL_0787, among the 15 highest abundant proteins in both proteomes) and lactate utilization proteins ABC (encoded by SMUL_1033–1035, in the top 40% abundant proteins) described to oxidize lactate in the related *Campylobacter jejuni*. The protein encoded by SMUL_0438 was detected in the proteomes of lactate/fumarate and pyruvate/fumarate-cultivated cells, but quantified to be only minor abundant and unsignificantly more abundant in lactate-cultivated cells (lowest 20% abundant proteins, Supplementary Data 4).

Cells adapted to pyruvate fermentation showed not many altered abundances of the aforementioned proteins involved in the fermentative physiology of *S. multivorans*. However, a molybdoenzyme encoded by SMUL_2101, was one of the ten most abundant during pyruvate fermentation (142-fold more abundant compared to fumarate-respiring cells). This noncharacterized molybdopterin oxidoreductase of the aldehyde/xanthine oxidoreductase family was also significantly more abundant (16.5-fold) in pyruvate fermentation-adapted *S. multivorans* cells (Table 2, Supplementary Data 5). A highly similar aldehyde oxidoreductase is also higher abundant during fermentation of *S. cavolei*. A BlastP query against the Genbank nr database revealed that orthologs of this protein are conserved in all *Sulfurospirillum* spp., but not in other Epsilonproteobacteria. Closely related proteins (>70% amino acid sequence identity over the whole sequence length) are encoded, among others, in several *Clostridium* spp. and in *Desulfovibrio* spp. known for their fermentative metabolism (~60% sequence identity). Additionally to this putative aldehyde oxidoreductase, three other proteins were observed to be more abundant in *S. multivorans* cells adapted to pyruvate fermentation and at the same time significantly more abundant in *S. multivorans* and *S. cavolei* cells during pyruvate fermentation. These include a membrane-integral cytochrome-like protein, a secreted type II asparaginase and a NosL family protein (Table 2, Supplementary Data 5).

**Syntrophy of *Sulfurospirillum multivorans* with *Methanococcus voltae*.** To unravel the potential role of *S. multivorans* in a syntrophic partnership as H$_2$ producer, a coculture with *M. voltae* was prepared. *M. voltae* is a methanogenic archaeon dependent on either H$_2$ or formate as electron donor and CO$_2$ as electron acceptor[40]. To investigate the syntrophic interaction of the two

organisms, the coculture was cultivated with lactate, which could not serve as a fermentation substrate for pure *S. multivorans* cultures. A syntrophic, hydrogen-consuming partner keeping H$_2$ concentration at a low level in cocultures might render lactate fermentation by *S. multivorans* thermodynamically feasible in a coculture. A medium optimized for *M. voltae* was used for the coculture (see Methods section). This medium contained a number of organic substances (i.e. the amino acids leucine and isoleucine, as well as casamino acids) not present in the medium of *S. multivorans* and several controls with *S. multivorans* pure cultures were performed to exclude any unprecedented growth effects potentially caused by this medium. The growth behavior of *S. multivorans* in the modified *M. voltae* medium with pyruvate alone as substrate was similar to that in the medium originally used for cultivation of *S. multivorans*, albeit a bit slower (Supplementary Figure 21A). The morphology of *S. multivorans* cells was unaltered in the *M. voltae* medium and independent from the type of cultivation—fermentatively or respiratory (Supplementary Figure 21B, C). The new medium with lactate as sole growth substrate could not promote growth for pure *S. multivorans* cultures. In the corresponding coculture, 15 mM lactate was consumed in approximately 2 weeks while methane was formed, indicating lactate fermentation by *S. multivorans* and H$_2$ transfer to *M. voltae* as syntrophic partner (Fig. 6a, b). To compare this result to a coculture with *S. multivorans* cells not adapted to fermentation, we cocultivated the nonadapted cells with *M. voltae* under otherwise identical conditions. Lactate was consumed in this coculture only slightly slower, taking about 20 days (Supplementary Figure 22). Electron microscopic analyses of the coculture revealed cell aggregates with sizes between 50 and 600 μm (Fig. 6c, Supplementary Figure 23). These aggregates showed a compact network of the rod-shaped *S. multivorans* and coccoidal *M. voltae* with net-forming flagellum-like structures surrounding the organisms. The cells in the aggregates were embedded in extracellular polymeric substances (EPS)-like structures, which might aid cell-to-cell contact.

## Discussion

In this study, production of H$_2$ was observed for several *Sulfurospirillum* species during pyruvate fermentation, which is the first evidence of fermentative H$_2$ production for Epsilonproteobacteria, which hitherto were generally regarded as H$_2$ oxidizers[18,34,41,42]. Specifically, we report H$_2$ production for *S. multivorans*, *S. cavolei*, *S. arsenophilum*, and *S. deleyianum* during fermentative growth on pyruvate. Sequential subcultivation on pyruvate alone revealed a continuous adaptation of *Sulfurospirillum* spp. to a fermentative metabolism. A comparison of the proteomes of cells adapted to pyruvate fermentation with nonadapted cells revealed four proteins which might have a positive effect on the growth rate during pyruvate fermentation of *S. multivorans*. The proposed role of these four proteins, namely an aldehyde oxidoreductase, a cytochrome, an asparaginase, and a NosL family protein, is discussed later in this chapter. However,

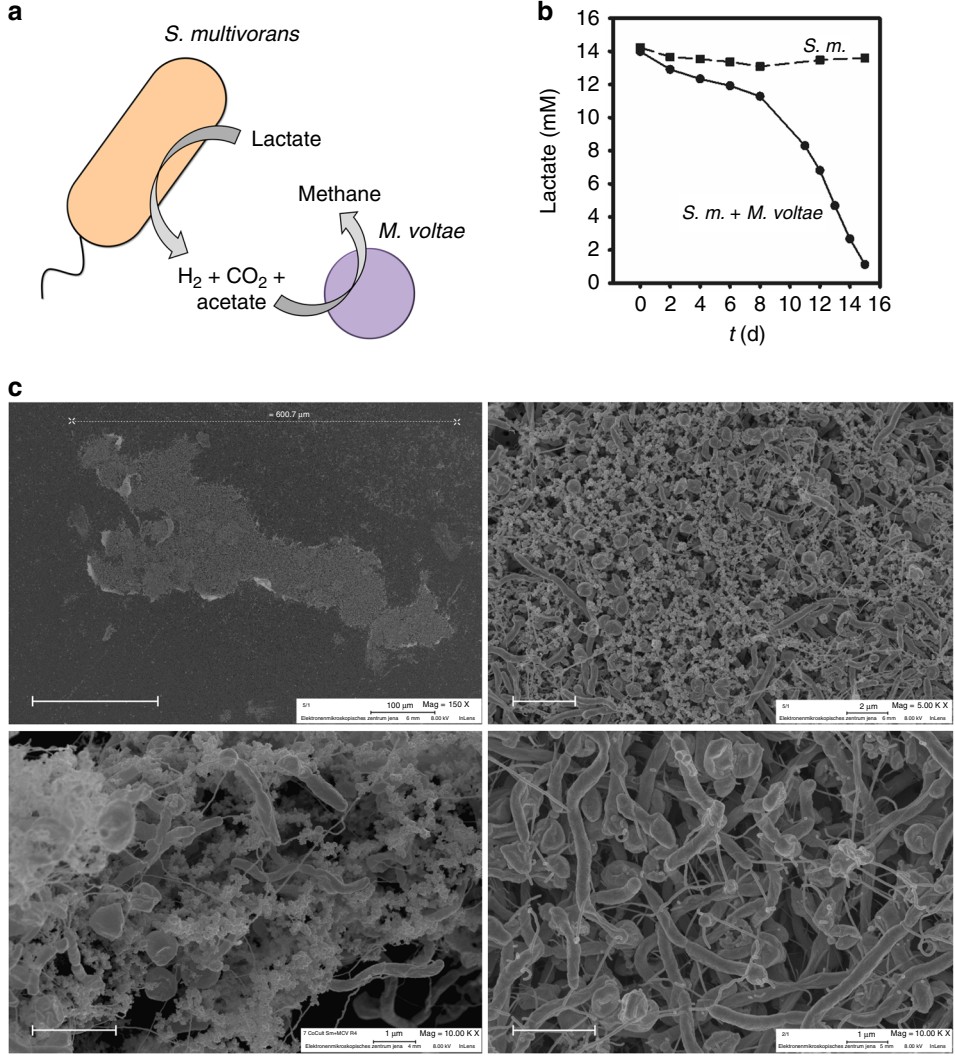

**Fig. 6** Syntrophic coculture of *S. multivorans* and *Methanococcus voltae*. **a** Scheme of syntrophic interactions and exchange of metabolites and **b** lactate concentration in *S. multivorans* pure culture and coculture of *S. multivorans* and *M. voltae*. **c** Electron microscopic images of aggregates, Magnifications, ×150 (whole aggregate, upper row left, scale bar 200 μm), ×5000 (upper right, scale bar 3 μm), ×10,000 (lower images, scale bar 2 μm). Sections of the lower images were obtained from different areas of the aggregate. White arrows indicate EPS-like structures. Cultivation experiments included three biological replicates in which similar aggregates were formed. S.m. *S. multivorans*. Pure *S. multivorans* and the coculture were cultivated in a medium originally optimized for *M. voltae* modified as described in the Methods section

the complete mechanism behind this long-term adaptation process in *Sulfurospirillum* spp. remains largely unresolved for now and might include also genomic rearrangements and/or population dynamics. An unresolved long-term regulatory effect similar to the one observed for *S. multivorans* after continuous transfer without PCE as electron acceptor[43] might play a role as well. The low adaptation to fermentative metabolism in *Sulfurospirillum* might also be an effect of the long-term respiratory cultivation of the organisms in our laboratory and cell culture collections. Whether *Sulfurospirillum* spp. in natural habitats shows an equally long adaptation time to fermentative metabolism is likely dependent on the type and concentration of electron donor and electron acceptor in the environment.

Two different fermentation balances were observed for the different *Sulfurospirillum* spp. tested. While *S. cavolei* showed the highest $H_2$ production rate and produced, besides hydrogen, acetate and $CO_2$, *S. deleyianum* and *S. multivorans*, displaying lower $H_2$ production, additionally produced succinate and lactate. Pyruvate is most likely oxidatively decarboxylated to acetyl-CoA by the pyruvate:ferredoxin oxidoreductase, which showed an

upregulation in the proteome of fermentatively cultivated compared to fumarate-respiring cells in both *S. multivorans* and *S. cavolei*. In contrast, the quinone-dependent pyruvate dehydrogenase (PoxB), which could transfer electrons generated upon pyruvate fermentation to menaquinone, is downregulated in fermenting cells and therefore most likely does not contribute significantly to pyruvate oxidation under this condition. A pyruvate formate lyase is not encoded in any *Sulfurospirillum* spp., which, in addition to the low protein abundance of a cytoplasmic formate dehydrogenase in *S. multivorans* and *S. cavolei*, argues against the role of the Hyf in an FHL complex as opposed to the suggested function for Hyf in *E. coli*[32]. The generated acetyl-CoA is used to generate acetate and one mol ATP per mol pyruvate via substrate-level phosphorylation.

The high abundance of an aldehyde oxidoreductase similar to the extensively studied molybdoenzyme of *D. gigas*[44] in the proteome of pyruvate-fermenting cells and especially those adapted to pyruvate fermentation points to the possible formation of an aldehyde as an intermediate during pyruvate fermentation. This aldehyde intermediate could be oxidized for energetic

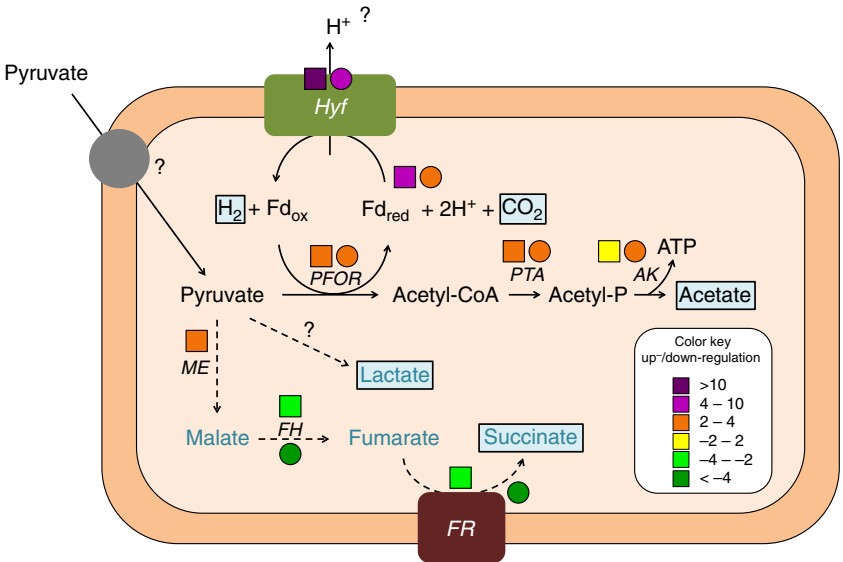

**Fig. 7** Tentative scheme of pyruvate fermentation metabolism in *S. multivorans* and *S. cavolei*. Reactions represented by solid arrows belong to the core pyruvate metabolism and are catalyzed by both organisms. Reactions with dashed arrows are solely catalyzed by *S. multivorans*, fumarate hydratase and fumarate reductase are also present in *S. cavolei*. Hyf might pump protons via its membrane-integral subunits (Supplementary Note 1, Supplementary Table 6 and Supplementary Figures 10–13) which could lead to additional ATP production via chemiosmotic coupling. Question marks indicate enzymes not identified. Fermentation products are highlighted in light blue boxes. Protein abundance ratios (pyruvate alone versus pyruvate/fumarate) are indicated by colored squares (*S. multivorans*) and circles (*S. cavolei*) at the protein abbreviations. Color code of the ratios is given in the box at the lower right. Hyf Hyf-like hydrogenase (SMUL_2383–2392; SCA02S_RS01920-RS01965), PFOR pyruvate:ferredoxin oxidoreductase (SMUL_2630; SCA02S_RS04525), PTA phosphotransacetylase (SMUL_1483; SCA02S_RS00245), AK acetate kinase (SMUL_1484; SCA02S_RS00240), ME malic enzyme (SMUL_3158; corresponding enzyme in *S. cavolei* is not present), FH fumarate hydratase (SMUL_1459, SMUL_1679–1680; SCA02S_RS00615-RS00620), FR fumarate reductase (SMUL_0550–0552; SCA02S_RS07735-RS07740)

reasons or for detoxification, which both could explain the faster growth after adaptation. However, aldehyde formation in *Sulfurospirillum* spp. is obscure. The most obvious source of aldehyde formation would be pyruvate decarboxylation by a pyruvate decarboxylase as observed in *Gluconobacter oxydans*[45]. Since such a decarboxylase is not encoded in the genome of any *Sulfurospirillum* spp., the corresponding reaction is unlikely to occur. Alternatively, acetaldehyde might be produced from acetate via acetyl-CoA and acetaldehyde oxidoreductase, possibly because of the high levels of acetate in the medium in the late phase of fermentation. Another enzyme also higher abundant in cells adapted to pyruvate is a type II asparaginase, which is secreted into the periplasm and suggested to play a role in generating the electron acceptor fumarate from L-asparagine via aspartate in anaerobically cultivated *E. coli* cells[46]. Exogenous asparagine was not present in the *S. multivorans* medium, but a general regulatory response to the lack of electron acceptors via a CRP-like regulator (detected in the proteome of *S. multivorans*, encoded by SMUL_3226) as suggested for *E. coli*[46] might be feasible. The role of the other two proteins higher abundant in pyruvate-adapted cells is more speculative. The cytochrome could play a role in electron transfer, while the NosL family protein, which is supposed to bind metal ions could act as a chaperone in the metal-locenter assembly of involved enzymes.

Electrons generated upon pyruvate oxidation are most likely transferred in both organisms to a ferredoxin of the *Allochromatium vinosum*-type, which is known for the very negative redox potentials of its two [4Fe4S] clusters[47]. The proteome data and biochemical experiments presented in our study strongly suggest that the Hyf (hydrogenase 4) of *Sulfurospirillum* spp. accepts electrons from the reduced ferredoxin to reduce two protons to hydrogen (Fig. 7). The *hyf* gene cluster is significantly upregulated,

whereas the other hydrogenases are either detected only in low amounts in the proteome data or are unaltered or downregulated under fermentative cultivation. Furthermore, reduced methyl viologen served as electron donor for $H_2$ production only with crude extract and not with intact cells, suggesting a cytoplasmic localization of the hydrogen-producing hydrogenase since methyl viologen should not have access to the cytoplasm. A cytoplasmic localization was also suggested previously for Hyf of *S. multivorans* based on the lack of a signal peptide in any of the corresponding subunit amino acid sequences[23,31]. The involvement of an Hyf in $H_2$ production via pyruvate oxidation was also observed for a group 4 hydrogenase from *Pyrococcus furiosus*[48] and a genetically modified *E. coli* strain[49]. The structure and subunit composition of several group 4 hydrogenases suggested their involvement in the generation of a proton motive force, thereby contributing to ATP formation[50,51]. A thorough alignment analysis of the subunits of *Sulfurospirillum* spp. Hyf indicated that most of the important residues in the membrane helices are conserved, thus making a role in energy conservation of this hydrogenase a possible scenario. The difference in the amount of $H_2$ produced with *S. cavolei* producing more $H_2$ than *S. multivorans* can be explained by two different fermentation metabolism types. Opposed to *S. cavolei*, reducing equivalents can be channelled into the production of lactate and succinate by *S. multivorans* (as was also observed for *S. deleyianum*) upon pyruvate fermentation. Succinate might be produced from fumarate (fumarate reductase) via malate (fumarase), which could be formed from pyruvate via reductive decarboxylation to malate by the malic enzyme (Fig. 7). This enzyme, which often functions in the reverse direction e.g. in $C_4$ plants, is upregulated in *S. multivorans* under fermentative conditions. This finding supports the involvement of the malic enzyme in conversion of pyruvate to malate. The malic enzyme was not detected in the proteomes of *S.*

*cavolei*, which might at least partially explain the different fermentation balances.

The origin of lactate in *S. multivorans* is not clear. A protein annotated as NAD+-dependent lactate dehydrogenase was not detected or was in low abundance in the proteomes and no NAD$(P)^+$-dependent lactate production could be measured. Most likely, the NAD(P)(H)-dependent lactate dehydrogenase is mis-annotated in the genome of *S. multivorans*, as reported for the corresponding protein of the related Epsilonproteobacterium *C. jejuni*[52]. A role of the *S. multivorans* "lactate dehydrogenase" in respiratory lactate oxidation was also unlikely as its low abundance in the proteome of lactate-grown cells suggest. Instead, lactate is likely oxidized in *S. multivorans* by orthologs of the NAD$^+$-independent enzymes recruited by *C. jejuni* for lactate oxidation, a flavin Fe-S cluster-containing enzyme and a three-partite lactate utilization protein[52]. These proteins were shown to be not substrate-inducible in *C. jejuni*, which is in line with the observed similar abundance of both proteins in lactate and pyruvate-cultivated *S. multivorans* cells. A possible source of lactate in pyruvate fermentation of *S. multivorans* could be the reduction of pyruvate via one of these or another NAD$^+$-independent lactate dehydrogenase (iLDH), which are mostly characterized to be functional in the direction of lactate oxidation[53,54]. In *Sulfurospirillum* spp., one could act in the reverse direction to produce lactate, possibly with reduced ferredoxin as electron donor. Of the several candidates of iLDHs of *S. multivorans*, only one of them shows a slight upregulation on pyruvate alone. A corresponding gene cluster is not encoded in the lactate-producing *S. deleyianum*, making it an unlikely candidate for lactate production. A glycolate oxidase was shown to be responsible for lactate oxidation in *Pseudomonas putida*[38,55] and a homolog is encoded in both lactate-producing *Sulfurospirillum* spp. This protein, however, is not upregulated upon pyruvate fermentation and further studies are needed to identify the lactate-producing enzyme in *S. multivorans*.

The different disposal of excess reducing equivalents during fermentation enables *S. multivorans* to grow with pyruvate even with 100% $H_2$ in the gas phase, whereas the growth of *S. cavolei* was nearly completely abolished under these conditions. This correlates with a shift towards a higher production of lactate and succinate and a lower acetate and $H_2$ production of *S. multivorans* under these conditions. $H_2$ production via Hyf is obviously subject to product inhibition and *S. multivorans* is able to circumvent this by using alternative cytoplasmic electron sinks upon fermentation.

The inability of *Sulfurospirillum* spp. to use lactate as sole substrate in pure cultures is most probably due to the thermodynamically unfavorable lactate oxidation to pyruvate upon $H_2$ production. To test the ecological significance of our observation, we established a lactate-consuming syntrophic partnership of *S. multivorans* with a hydrogen-consumer, *M. voltae*. Indeed, this coculture enabled lactate utilization by *S. multivorans* and the formation of large cell aggregates of the two organisms presumably via the formation of EPS was observed. Since *M. voltae* is not able to thrive in minimal medium[40], we cultivated the coculture not in the medium used for *S. multivorans* cultivation, but in a medium for *M. voltae* which included several organic substances. The influence of this medium on *S. multivorans* cultivation was negligible, so that the observed lactate degradation and aggregate formation can be unequivocally linked to the syntrophic coculture. The interspecies electron carrier in the syntrophic relationship of *S. multivorans* and *M. voltae* is hydrogen based on the metabolite analysis of pure *S. multivorans* cultures, where only hydrogen was found as a product of pyruvate fermentation. However, lower levels of formate (<0.5 mM) could not be detected in our analysis and therefore we cannot

completely exclude formate as a second electron carrier of lower importance here. Production of formate from $CO_2$ in *S. multivorans* could be mediated by a cytoplasmic formate dehydrogenase (SMUL_0079), for which the electron donor and catalytic bias is yet unknown. The three periplasmic formate dehydrogenases of *S. multivorans* are putatively formate-oxidizing and most probably connected to the menaquinone pool via a cytochrome *b* subunit and therefore the production of formate by these enzymes would be thermodynamically unfavorable, since menaquinones have a relatively positive midpoint potential of around −100 mV. Direct interspecies electron transfer is also unlikely to proceed in this syntrophic interaction, since besides none of the typical nanowire structures were visible in electron microscopy, no pili or extracellular cytochromes[56], usually used for interspecies electron transfer of e.g. *Shewanella* spp. and *Geobacter* spp.[57,58], were found in the genome of any *Sulfurospirillum* spp. Both *S. multivorans* cells adapted and not adapted to pyruvate fermentation supported the growth of the coculture, which strengthens our suggested role of *Sulfurospirillum* spp. as $H_2$ producers in anaerobic food webs. Additionally, this role as a potential $H_2$ producer is most likely not limited to this genus. In a genome mining approach, *hyf* gene clusters were found among several genera of Epsilonproteobacteria inhabiting a wide range of habitats. Some *Campylobacter* spp. known to be opportunistic or food-borne pathogens encode the same Hyf as *Sulfurospirillum* spp., while *hyf* gene clusters containing either a formate channel gene in different *Campylobacter* spp. or additionally a cytoplasmic formate dehydrogenase in other phyla might indicate the formation of an FHL complex. Since a PFL is missing in these bacteria, it might be presumed that extracellular formate might aid growth in these bacteria as reported for *Thermococcus* spp.[59]. Some *Sulfurospirillum* spp. even encode for both an FHL-independent Hyf and one presumably forming an FHL complex, pointing towards separate regulation and roles of both hydrogenases and thus for even more physiological diversity in this genus.

Taken together, our results show that several Epsilonproteobacteria have to be considered as $H_2$ producers and serve as syntrophic partners in e.g. the presence of lactate, which is a widely distributed organic electron donor in natural habitats. $H_2$ production in *Sulfurospirillum* spp. under the tested conditions relies on Hyf, a multisubunit, membrane-bound and cytoplasmically oriented group 4 NiFe hydrogenase similar to the one used in a second *E. coli* FHL complex and probably functioning as a proton pump. Adaptation to fermentative conditions seems to be common in *S. multivorans* and related strains, although the underlying mechanism of this process is still unclear. Two separate clades of *Sulfurospirillum* spp. have different fermentation pathways, the *S. cavolei* clade producing more $H_2$ and exclusively one organic acid, namely acetate, in comparison to *S. multivorans*, which additionally produces lactate and succinate. All these findings imply an even higher versatility for Epsilonproteobacteria than previously thought, although it should be noted here that most of them primarily rely on a respiratory metabolism. Still, our established coculture with a methanogen suggests a new ecological role for *Sulfurospirillum* spp., which inhabit a large range of environmentally or biotechnologically important habitats such as wastewater plants, oil reservoirs, bioelectrodes, contaminated sediments or marine areas.

## Methods

**Cultivation of bacteria**. *S. multivorans* was cultivated under anaerobic conditions at 28 °C in a defined mineral medium without vitamin $B_{12}$ (cyanocobalamin) as follows. The basal medium contained per L: 70 mg $Na_2SO_4$, 200 mg $KH_2PO_4$, 250 mg $NH_4Cl$, 1 g NaCl, 400 mg $MgCl_2 \cdot 6 H_2O$, 500 mg KCl, and 150 mg $CaCl_2 \cdot 2 H_2O$. Medium was made anoxic (under $N_2$ (150 kPa)) and autoclaved and

completed with anoxic (under $CO_2$ atmosphere) and autoclaved 40 mL $NaHCO_3$ solution (84 g $L^{-1}$), 2.5 mL iron sulfate solution ($FeSO_4 \cdot 7 H_2O$, 9.2 g $L^{-1}$, dissolved in 50 mM (v/v) $H_2SO_4$), 1 mL cysteine solution (cysteine-HCl, 50 g $L^{-1}$) and 10 mL supplement solution, each of them made anoxic under $N_2$ and autoclaved. The supplement contained per L: 100 mL potassium phosphate buffer ($KH_2PO_4$, 25.3 g $L^{-1}$; $K_2HPO_4$, 141.8 g $L^{-1}$; pH 7.5). The following anoxic, sterile solutions were added via a sterile filter (0.2 µm pore size): 20 mL trace element solution (see below), 10 mL vitamin solution (see below), 2 mL selenite solution ($Na_2SeO_3 \cdot 5 H_2O$, 130 mg $L^{-1}$) and 4 mL tungstate solution ($Na_2WO_4 \cdot 2 H_2O$, 165 mg $L^{-1}$). The trace element solution contained per L 10 mL HCl (25% (w/v)), 1 g $FeSO_4 \cdot 7 H_2O$, 70 mg $ZnCl_2$, 100 mg $MnCl_2 \cdot 4 H_2O$, 6 mg $H_3BO_3$, 190 mg $CoCl_2 \cdot 6 H_2O$, 2 mg $CuCl_2 \cdot 2 H_2O$, 24 mg $NiCl_2 \cdot 6 H_2O$, 36 mg $Na_2MoO_4 \cdot 2 H_2O$. The vitamin solution contained per L 0.1 g D(+)biotin, 0.4 g 4-aminobenzoic acid, 1 g nicotinic acid, 0.5 g Ca-D(+) pantothenate, 1.5 g pyridoxamine-2HCl, 1 g thiamine-HCl.

Pyruvate or lactate (40 mM) were used as electron donor and fumarate (40 mM) as electron acceptor. For fermentation experiments, all cultivations were performed with pyruvate (40 mM) or lactate (40 mM) as sole energy source in the absence of an electron acceptor and without yeast extract. Bacteria were grown in serum bottles with a ratio of aqueous to gas phase of 1:1. If not stated otherwise, the gas phase was $N_2$ (150 kPa). For the cultivation with 100% $H_2$ in the gas phase, nitrogen was completely removed after autoclaving by flushing with $H_2$ and an overpressure of 50 kPa was applied. Fermentation balance experiments were performed at 28 °C in 1 L Schott bottles placed in a fermentation apparatus to allow for the expansion of the gases during the cultivation and to determine the stoichiometry of dissolved and gaseous fermentation products (Supplementary Figure 2). For $CO_2$ quantification, the gas phase of the Schott bottle was connected via a tube to a washing flask filled with 200 mL 4 M KOH to bind produced $CO_2$ as carbonate. Downstream, the gas phase of the washing flask was further connected to a water-filled measuring cylinder placed upside down in a water bath. The amount of $H_2$ was determined volumetrically via the displaced volume of water in the measuring cylinder that correlates with the amount of $H_2$ produced. The gas volume was adjusted to standard temperature (25 °C) and pressure (1 bar). The concentration was calculated using the ideal gas equation. The adaptation experiment included a transfer in the next subcultivation step every 48 h with 10% inoculum in the S. multivorans medium as stated above. Comparison of the proteome profiles during the adaptation process was performed with cells harvested after the third transfer on pyruvate alone and with cells completely adapted to pyruvate fermentation over more than 25 subcultivation steps. Clostridium pasteurianum was cultivated in anoxic media composed of 1 L basal medium (autoclaved) supplemented with the following anoxic solutions: 100 mL phosphate buffer (142 g $L^{-1}$ $K_2HPO_4$, 15 g $L^{-1}$ $KH_2PO_4$) and 5 mL iron solution (10 g $L^{-1}$ $FeSO_4 \cdot 7 H_2O$). The basal medium contained per L 142 mg NaCl, 1.42 g $NH_4Cl$, 284 mg $MgSO_4 \cdot 7 H_2O$, 14.2 mg $Na_2MoO_4 \cdot 2 H_2O$, 28.4 mg D(+) biotin and 1.42 mg 4-aminobenzoate. Cells were grown in rubber-stoppered serum bottles with a ratio of aqueous to gas phase of 1:4. pyruvate (40 mM) and glucose (20 mM) were used as substrates. Desulfitobacterium hafniense DCB-2 was cultivated anoxically on medium containing 70 mg $Na_2SO_4$, 200 mg $KH_2PO_4$, 250 mg $NH_4Cl$, 1000 mg NaCl, 400 mg $MgCl_2 \cdot 6 H_2O$, 500 mg KCl, and 150 mg $CaCl_2 \cdot 2 H_2O$, and 2 g yeast extract per L, supplemented with the following anoxic and filter-sterilized solutions: 0.5 mL vitamin solution (80 mg 4-aminobenzoic acid, 20 mg D-(+) biotin, 200 mg nicotinic acid, 100 mg Ca-D-(+)pantothenate, 300 mg $L^{-1}$ pyridoxamine·2 HCl, and 200 mg thiamine·HCl per L), 2 mL trace element solution SL10, 1 mL B12 solution (cyanocobalamin, 50 mg $L^{-1}$), 0.2 mL selenite solution ($Na_2SeO_3 \cdot 5 H_2O$, 26 mg $L^{-1}$), 0.1 mL tungstate solution ($Na_2WO_4 \cdot 2 H_2O$, 33 mg $L^{-1}$), 30 mL $NaHCO_3$ solution (84 g $L^{-1}$) (autoclaved separately under $CO_2$) and 10 mL cysteine solution (cysteine·HCl, 50 g $L^{-1}$) (autoclaved separately under $N_2$). The final pH of the medium was between 7.4 and 7.8. The S. multivorans/M. voltae cocultures were maintained in modified medium according to Whitman et al.[40]. Cocultures were grown in rubber-stoppered serum bottles with a ratio of aqueous to gas phase of 1:1. The gas phase was $N_2/CO_2$ (80:20 (v/v); 150 kPa) and lactate (15 mM) the electron donor. Syntrophic cocultures were initially inoculated in a ratio of Methanococcus to Sulfurospirillum cells of 5:1. Subculturing was done by transferring 10% (v/v) of the coculture into freshly prepared media for ten subcultures before the growth experiment and electron microscopy were performed. The basal medium contained per L 0.1 g isoleucine, 0.1 g leucine, 0.34 g KCl, 4 g $MgCl_2 \cdot 6 H_2O$, 3.45 g $MgSO_4 \cdot 7 H_2O$, 0.25 g $NH_4Cl$, 0.14 g $CaCl_2 \cdot 2 H_2O$, 0.14 g $K_2HPO_4$, 5 g NaCl. After boiling and cooling to room temperature under $N_2$, following separately autoclaved anoxic solutions were added (per L basal medium): 10 mL vitamin solution, 10 mL trace element solution (see below), 0.2% (w/v) Casamino Acids (Difco Laboratories, Detroit, Michigan, USA), 10 mL sulfide solution ($Na_2S \cdot 9 H_2O$, 50 g $L^{-1}$) and 10 mL cysteine solution (L-cysteine-HCl, 50 g $L^{-1}$). Final pH was adjusted to 7.0 with 2.5 g $L^{-1}$ $NaHCO_3$. The vitamin solution contained per L 2 mg D(+) biotin, 2 mg folic acid, 10 mg pyridoxamine-2HCl, 5 mg thiamine-HCl · 2 $H_2O$, 5 mg riboflavin, 5 mg nicotinic acid, 5 mg Ca-D(+) pantothenate, 0.1 mg vitamin $B_{12}$, 5 mg 4-aminobenzoic acid, 5 mg lipoic acid. The trace element solution contained per L 1.5 g nitriloacetic acid (firstly added, and pH adjustment to 6.5 with KOH), 3 g $MgSO_4 \cdot 7 H_2O$, 0.5 g $MnSO_4 \cdot H_2O$, 0.1 g $FeSO_4 \cdot 7 H_2O$, 0.18 g $CoSO_4 \cdot 7 H_2O$, 0.1 g $CaCl_2 \cdot 2 H_2O$, 0.18 g $ZnSO_4 \cdot 7 H_2O$, 0.01 g $CuSO_4 \cdot 5 H_2O$, 0.02 g $KAl(SO_4)_2 \cdot 12 H_2O$, 0.01 g $H_3BO_3$, 0.01 g $Na_2MoO_4 \cdot 2 H_2O$, 0.03 g $NiCl_2 \cdot 6 H_2O$, 0.3 mg $Na_2SeO_3 \cdot 5 H_2O$, 0.4 mg $Na_2WO_4 \cdot 2 H_2O$. All Sulfurospirillum spp. (S. multivorans—DSM 12446, S. cavolei type strain Phe 91

(NBRC 109482)—DSM 18149, S. halorespirans—DSM 13726, S. barnesii—DSM 10660, S. deleyianum—DSM 6946, S. arsenophilum DSM 10659), C. pasteurianum (DSM 525), D. hafniense DCB-2 (DSM 10664), and E. coli JM109 (DSM 3423) and M. voltae (DSM 1537) were initially obtained from the German Collection of Microorganism (DSMZ, Braunschweig, Germany).

**Cell harvesting and fractionation.** S. multivorans, S. cavolei, and C. pasteurianum W5 cells were harvested in the mid-exponential growth phase in an anoxic glove box (COY, 134 Laboratory, Grass Lake, Michigan, USA) by centrifugation ($12,000 \times g$, 10 min at 10 °C). For the preparation of cell suspensions, the obtained cell pellets were washed twice in anoxic 100 mM MOPS-KOH-buffer (pH 7.0) and resuspended in two volumes (2 mL per g cells) of the same buffer. Subcellular fractionation was done by washing the cell pellet twice in 50 mM Tris-HCl (pH 8.0) and resuspension (2 mL per g cells) in the same buffer containing DNase I (AppliChem, Darmstadt, Germany) and protease inhibitor (one tablet for 10 mL buffer; complete Mini, EDTA-free; Roche, Mannheim, Germany). The resuspended cells were disrupted using a beadmill (10 min at 25 Hz; MixerMill MM400, Retsch GmbH, Haan, Germany) with an equal volume of glass beads (0.25–0.5 mm diameter, Carl Roth GmbH, Karlsruhe, Germany). The crude extracts were separated from the glass beads by centrifugation ($14,000 \times g$, 2 min) under anaerobic conditions and ultracentrifuged ($36,000 \times g$, 45 min at 4 °C). The obtained supernatants were considered as soluble fractions (SF). The pellets were washed twice with 50 mM Tris-HCl (pH 8.0) including protease inhibitor (one tablet for 10 mL buffer; cOmplete Mini, EDTA-free; Roche, Mannheim, Germany) and resuspended in the same buffer. The suspension was stated as membrane fraction (MF).

**Measurement of hydrogenase activity.** $H_2$-oxidizing activity was measured in $H_2$-saturated buffer (50 mM Tris-HCl, pH 8.0) with 1 mM benzyl viologen (BV) or methyl viologen (MV) at 30 °C as artificial electron acceptors. The reduction of the redox dyes was followed at 578 nm using a Cary 100 spectrophotometer (Agilent Technologies, Waldbronn, Germany). $H_2$-evolving activities of cell extracts were determined gas chromatographically with 1 mM MV as electron donor: MV was reduced with 20 mM sodium dithionite in an anoxic buffer system (50 mM Tris-HCl, pH 8.0). Protein concentration was determined according to the method of Bradford[60]. Hydrogenase enzyme activities are given in nanokatal units (1 nmol $H_2$ evolved per second).

**Analytical methods.** Liquid samples were taken anaerobically, filtered with 0.2 µm syringe filters (MiniSart RC4, Sartorius, Göttingen, Germany) and acidified with concentrated $H_2SO_4$ (2.5 µL $mL^{-1}$ sample volume). Organic acids were separated by HPLC at 50 °C on an AMINEX HPX-87H column (7.8 × 300 mm, Bio-Rad, Munich, Germany) with a cation H guard pre-column using 5 mM $H_2SO_4$ as mobile phase at a flow rate of 0.7 mL $min^{-1}$. The injection volume was 20 µL per sample. All acids (e.g. pyruvate, acetate, lactate, succinate, and fumarate) were monitored by their absorption at 210 nm. Retention times were compared to known standards and concentrations were calculated using calibration curves. $H_2$ was measured gas chromatographically with 99.999% argon as the carrier gas using a thermal conductivity detector (AutoSystem, Perkin Elmer, Berlin, Germany). Samples for gas analysis were taken from the gas phase with gas-tight syringes (Hamilton, Bonaduz, Switzerland). Concentrations were calculated using calibration curves. $CO_2$ formed during the cultivation was determined gravimetrically. To 15 mL of the solution of the $CO_2$ trap 7.5 mL $NH_4Cl$ (1 M) and 15 mL $BaCl_2$ (1 M) were added and the pH was adjusted to 9 with concentrated HCl (37%). After stirring for 2 h at room temperature, the precipitated barium carbonate was filtered with filter circles and dried overnight at 80 °C.

**Reverse transcription and polymerase chain reaction.** Total RNA from three independent S. cavolei cultures was isolated from cells in the mid-exponential growth phase using the RNeasy minikit (Qiagen, Hilden, Germany). Residual genomic DNA (gDNA) in the RNA samples was removed with DNase I (RNase free; Roche, Mannheim, Germany). RNA quality was checked by visual inspection after agarose gel electrophoresis using distinct rRNA bands as control. Synthesis of cDNA was done with 1 µg RNA as starting material in the RevertAid First Strand cDNA Synthesis kit (Thermo Scientific, Schwerte, Germany). The RT-PCE mixture contained 1 µg RNA, 2.5 µL reverse primer, 2 µL 10 mM dNTP mix and 3.5 µL 5× reaction buffer and PCR-grade water (Fermentas, St. Leon Rot, Germany) was added to a final volume of 17.5 µL. ReverseAid Reverse transcriptase (RT, 0.5 µL, 200 U $mL^{-1}$) was added to 10.5 µL of the mixture (positive control), residual 7 µL of the reaction mixture without RT was used as negative control. Reaction mixtures were incubated for 1 h at 42 °C, the reaction was stopped for 5 min at 72 °C. PCR was performed with each 1 µL of positive and negative reactions, 2.5 µL forward and reverse primer, 1 µL 10 mM dNTP mix, 2.5 µL 10× reaction buffer, 14.5 µL PCR-grade water (Fermentas, St. Leon Rot, Germany) and 1 µL Taq polymerase (0.1 U $µL^{-1}$, Thermo Scientific, Schwerte, Germany) in a thermocycler (Mastercycler, Personal, Eppendorf, Hamburg) with the following program: 95 °C for 5 min, 30 cycles of 95 °C for 1 min, 52 °C for 30 s, 72 °C for 1 min and final elongation at 72 °C for 10 min. S. halorespirans gDNA was isolated using the innuPREP Bacteria DNA kit (Analytik Jena AG, Jena, Germany) according to the manufacturer's instructions. Quality of the extracted DNA was confirmed by gel

electrophoresis. The PCR reaction mixture contained 1 µg DNA, 2.5 µL forward and reverse primer, 1.5 µL 10 mM dNTP mix, 5 µL HF reaction buffer and 0.5 µL Phusion DNA polymerase (2 U µL$^{-1}$, Thermo Scientific, Schwerte, Germany). The mix was filled up to 25 µL with PCR-grade water (Fermentas, St. Leon Rot, Germany). The PCR program included following steps: 96 °C for 5 min, 30 cycles of 96 °C for 1 min, 60 °C for 30 s, 72 °C for 30 s and final elongation at 72 °C for 10 min in a thermo cycler (Mastercycler, Personal, Eppendorf, Hamburg). Used primer pairs are listed in Supplementary Table 4.

**Field emission-scanning electron microscopy.** Field emission-scanning electron microscopy (FE-SEM) was performed with cocultures of *S. multivorans* and *M. voltae*. After incubation of 3 mL culture in 2.5% glutaraldehyde for 15 min, the cells were pre-fixed for 2 h on poly-L-lysin coated cover slides (12 mm, Fisher Scientific, Schwerte, Germany). Washing of cover slides was done using 0.1 M sodium cacodylate (pH 7.2) (>98% purity, Sigma Aldrich, Steinheim, Germany) for three times. Subsequently, cells were post-fixed with 1% osmium tetroxide in the same cacodylate buffer and dehydrated with different ethanol concentrations. Critical point drying was done in a Leica EM CPD200 Automated Critical Point Dryer (Leica, Wetzlar, Germany) and the samples were coated with 6 nm platinum in a BAL-TEC MED 020 Sputter Coating System (BAL-TEC, Balzers, Liechtenstein). They were visualized at different magnifications using a Zeiss-LEO 1530 Gemini field emission-scanning electron microscope (Carl Zeiss, Oberkochen, Germany).

**Protein sample preparation and proteomics.** Protein concentration of extracted proteins was determined using a Bradford reagent (Bio-Rad, Munich, Germany) with bovine serum albumin as standard. For protein identifications, 20 µg of crude extracts were first cleaned from cations and cell debris by running shortly into an SDS gel. For this, the gel was run at 13 mA until the proteins entered the separating gel at a depth of about 3–5 mm. Then the protein band was cut out, reduced, alkylated and proteolytically digested with trypsin (Promega, Madison, WI, USA) and subsequently desalted and concentrated with C18 ZipTip pipette tips (Merck Millipore).

Mass spectrometry was performed using an Orbitrap Fusion (Thermo Fisher Scientific, Waltham, MA, USA) coupled to a TriVersa NanoMate (Advion, Ltd., Harlow, UK). 5 µL of the peptide solution was separated with a Dionex Ultimate 3000 nano-LC system (Dionex/Thermo Fisher Scientific, Idstein, Germany) using a 15 cm analytical column (Acclaim PepMap RSLC, 2 µm C18 particles, Thermo Scientific) at 35 °C. Liquid chromatography was done with a constant flow of 300 nL min$^{-1}$ with a mixture of solvent A (0.1% formic acid) and B (80% acetonitrile, 0.08% formic acid) in a linear 90 min gradient of 4 to 55% solvent B.

MS1 scans were taken with a cycle time of 3 s in the Orbitrap mass analyzer between 350 and 2000 *m/z* at a resolution of 120,000, automatic gain control (AGC) target $4\times10^5$, maximum injection time 50 ms. Data-dependent acquisition was employed selecting for highly intense ions (>$5\times10^4$) and charge state between +2 and +7 with a precursor ion isolation windows of 1.6 *m/z*. Fragmentation was done via higher energy dissociation at 30% energy, and also measured in the Orbitrap analyzer at a resolution of 120,000 with an AGC target of $5\times10^4$ and a maximum injection time of 120 ms. Fragmentation events were done within the 3 s of cycle time until the next MS1 scan was done excluding the same mass (±10 ppm) for further precursor selection for 45 s.

Mass spectrometric data were analyzed with Proteome Discoverer 1.4 (pyruvate fermentation and fumarate respiration of *S. multivorans* and *S. cavolei*) or 2.2 (lactate and pyruvate oxidation with fumarate as electron acceptor and pyruvate fermentation adaptation, both Thermo Scientific) against the NCBI *S. multivorans* (CP007201.1) or *S. cavolei* (BBQE00000000.1) database with the search engines SequestHT and MS Amanda. Oxidation of methionine was set as dynamic, carbamidomethylation of cysteine as static modification; two missed cleavages were accepted, mass tolerance of MS1 and MS2 measurements were set to 5 ppm and 0.05 Da, respectively. For the lactate/fumarate versus pyruvate/fumarate and pyruvate adaptation experiments, a quantification across samples with Minora and Precursor ions quantifier was performed, which allows identification of peptides with a hit exclusively in MS1 measurements when peptides appeared at the same retention time across all samples. A percolator false discovery rate threshold of <0.01 was set for peptide identification. Label-free quantification of proteins was done with the area of the three most abundant peptides of each protein. The values were logarithmized (log10) and normalized (see Supplementary Data 1 and 3) and a two-tailed *t* test was applied using prostar proteomics[61]. Significance values (*p* values) of <0.05 were considered to indicate statistical significance. Only proteins identified in at least two of the three replicates were quantified; otherwise, proteins were considered to be identified but were not quantified.

## Data availability

The raw proteomic data have been deposited in the PRIDE repository with the accession numbers PXD010316 (*S. multivorans* and *S. cavolei* pyruvate fermentation) and PXD010303 (pyruvate fermentation adaptation and lactate oxidation).

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

## Acknowledgements

This work was funded by the German Research Council (DFG)—Jena School for Microbial Communication (JSMC) and Research Unit FOR1530. We would like to gratefully acknowledge Susanne Linde (University Hospital Jena, Center for Electron Microscopy) for the field emission-scanning electron microscopic analysis. The presented work included the use of analytical facilities of the Centre for Chemical Microscopy (ProVIS) at the Helmholtz Centre for Environmental Research (UFZ Leipzig). ProVIS is funded by the European Regional Development Funds (EFRE—Europe funds Saxony) and the Helmholtz Association. The authors would like to thank Benjamin Scheer (UFZ Leipzig) for invaluable assistance in the lab with mass spectrometry and Dominique Türkowsky (UFZ Leipzig) for help with statistical analysis of proteome data.

## Author contributions

S.K. performed the wetlab work, S.K. and T.G. planned experiments, T.G. initiated the study, T.G. and G.D. supervised the study, L.A. performed the mass spectrometric analysis, S.K., T.G., and G.D. analyzed and discussed the data, M.W. was responsible for electron microscopy, S.K. and T.G. drafted the manuscript, all authors revised, read and approved this manuscript.

## Additional information

**Competing interests:** The authors declare no competing interests.

