## [Peer Review File · Nature Communications]

Reviewers' comments:

Reviewer #1 (Remarks to the Author):

The manuscript by Kruse et al describes hydrogen production by *Sulfurospirillum* spp. and their ability to grow in syntrophic association with methanogens.

This study makes an important contribution to science, especially as hydrogen production and the ability to grow in syntrophic association with methanogens thus far is a property associated with bacteria belonging to the deltaproteobacteria and the Firmicutes. When this ability can be unequivocally linked to other phylogenetic groups, this is an important step forward. The manuscript contains a set of solid experiments, but in my opinion some extra experiments and editorial changes are needed to make the story more complete and conclusive.

The following main points are for the authors to consider:

The introduction gives the impression that hydrogen is the only interspecies electron transfer compound in syntrophic methanogenic communities. Formate transfer and direct electron transfer need to be mentioned in the introduction.

The study includes proteome experiments of cells grown with pyruvate and cells grown with pyruvate and the electron acceptor fumarate. Why were no proteome experiments done with cells in different stages of adaptation of growth on pyruvate. As growth of *S. multivorans* on pyruvate becomes faster upon subculturing, proteome or transcriptome analysis would be an ideal way to get insight into what is changing.

In the manuscript two media are mentioned, one for the pure culture experiments and one for the coculture experiments. The medium for *S. multivorans* (ref 30) is essentially different from the *M. voltae* medium (ref 32). The latter medium contains a large number of organic compounds, which needs to be considered in the experimentation and in the interpretation of the results. By the way, in the publication of Whtman et al different media are presented; it is not clear which of these medium was used in the current study. From the description it is also not clear if both the pure culture and the coculture experiment described in Figure 6 were done in the *M. voltae* medium. It is poorly described how these cocultures were constructed and maintained. What is the inoculum size of each and how is subculturing procedure of the coculture? What are the growth properties of a pure culture of *S. multivorans* in the *M. voltae* medium? Is the observed aggregate formation caused by syntrophic growth or by the composition of the medium? This needs further description and clarification.

The syntrophic behaviour is just discussed as a result of interspecies hydrogen transfer. The option of interspecies formate transfer or direct electron transfer is ignored. This is a shortcoming, especially because *M. voltae* is able to use both hydrogen and formate. The authors mention that the bacterium does not have a pyruvate formate lyase, and that this would make a role of formate less likely. However, formate can also be formed from electrons derived from lactate oxidation and pyruvate oxidation. A proteome or transcriptome analysis of cells of *S. multivorans* grown in pure culture with pyruvate, lactate with fumarate and in coculture with lactate is recommended. Such an experiment would avoid to draw conclusions that are biased. It would also allow to get insight into which enzyme is (enzymes are) involved in lactate oxidation to pyruvate. When lactate would be oxidized by an NAD-dependent enzyme (SMUL_0438), it is also not clear how NADH is oxidized coupled to hydrogen formation. The possible involvement of formate dehydrogenase (Smul-0438) should be considered.

Hydrogenase activity measurements were done with methylviologen and benzylviologen. Is there any activity with NADH or NAD in the hydrogen formation and hydrogen oxidation reaction, respectively. Was it possible to detect formate dehydrogenase activity in cell extracts?

Minor point:

In the introduction it is mentioned that hydrogen production was never shown for Epsilonbacteria. This statement is not completely true. Hydrogen production was shown for *Sulfurospirillum carboxydovorans* (Jensen and Finster (2005) *Antonie van Leeuwenhoek* 87: 339–353. This is reference 28 of the manuscript.

Reviewer #2 (Remarks to the Author):

The authors show that some *Sulfurospirillum* spp., which had previously been viewed only as hydrogen oxidizers, produce hydrogen during pyruvate fermentation. In addition, *S. multivorans* was shown to grow syntrophically on lactate when a hydrogenotrophic methanogen, *Methanococcus voltae*, was present. Enzyme assays reveal that hydrogen-producing activity was present in the membrane fraction and likely cytoplasmically oriented. Hydrogen production activity was correlated to hydrogenase 4 (Hyf) based on proteomic and transcriptional analyses. Hyf gene cluster was detected in genomes of other *Sulfurospirillum* and in some *Campylobacter* species.

Interestingly, it took many transfers for *Sulfurospirillum* spp. to ferment pyruvate and produce hydrogen effectively. Some *Sulfurospirillum* spp. produce mainly acetate, hydrogen, and carbon dioxide from pyruvate while other produce less hydrogen due to their ability to produce fermentation products such as succinate. The work on fermentation products was done well but the results are not surprising.

Specific comments:

1. The authors concluded that some members of the Epsilonproteobacteria should be considered to be hydrogen producers in their natural habitats. It is likely that some will grow syntrophically on lactate as shown for *S. multivorans*. However, the ecological importance of hydrogen production from pyruvate is not clear. It took many transfers in the laboratory before *Sulfurospirillum* spp. effectively fermented pyruvate and produced hydrogen. This argues that the ecological role of these organisms is not pyruvate fermentation. In general, *Sulfurospirillum* spp. do not oxidize or ferment carbohydrates so fermentative hydrogen production from pyruvate derived from sugars is not likely.
2. The authors provided proteomic data in supplemental tables excel files, but these data should also be deposited in a database such as PRIDE.

Minor comments:

1. p. 33. Spell out genus name here.
2. Line 109. I would state that the gas volume was adjusted to standard temperature and pressure.
3. Line 110. Was medium used for the adaptation experiment the same as that described on lines 94-95?
4. Line 181. manufacturer's
5. Line 231. There is no period at the end of the sentence making me wonder if there is some text missing.
6. Lines 244-246. This statement seems out of place as the data are discussed in the next section.
7. Lines 259-261. Cite Fig. 2
8. Line 330-331, periplasmically oriented and cytoplasmically oriented would be better wording, as both are located in the cytoplasmic membrane.
9. Line 350-445. Fig. 5 needs to be cited. The order of topics discussed could be improved. I suggest discussion of the data on hydrogenases together with the data on Hyf first (lines 385-403) to make it clear that Hyf is the one involved in hydrogen production. One can then discuss pyruvate/energy metabolism and anabolism.

10. Lines 400-403. The authors state that an analysis of the amino acid involved in proton translocation was done but they do not state what they conclude from these data.

Michael J. McInerney

Reviewers' comments:

Reviewer #1 (Remarks to the Author):

The manuscript by Kruse et al describes hydrogen production by *Sulfurospirillum* spp. and their ability to grow in syntrophic association with methanogens.

This study makes an important contribution to science, especially as hydrogen production and the ability to grow in syntrophic association with methanogens thus far is a property associated with bacteria belonging to the deltaproteobacteria and the Firmicutes. When this ability can be unequivocally linked to other phylogenetic groups, this is an important step forward. The manuscript contains a set of solid experiments, but in my opinion some extra experiments and editorial changes are needed to make the story more complete and conclusive.

--- We thank the reviewer very much for seeing the importance in our work! We performed several extra experiments in turn to your constructive criticism and we think that the manuscript is strongly enhanced after revision. We just could not follow several of the reviewers points on formate, since we never, under any condition, detected formate in our experiments. However we added some discussion on that as well (see below).

The following main points are for the authors to consider:

The introduction gives the impression that hydrogen is the only interspecies electron transfer compound in syntrophic methanogenic communities. Formate transfer and direct electron transfer need to be mentioned in the introduction.

--- We now added "Besides H₂, also formate, similarly formed during fermentative metabolism, is an important electron carrier in e.g. syntrophic fatty acid-degrading methanogenic consortia⁵⁹.", and mentioned H₂ AND formate at the other points in the introduction.

The study includes proteome experiments of cells grown with pyruvate and cells grown with pyruvate and the electron acceptor fumarate. Why were no proteome experiments done with cells in different stages of adaptation of growth on pyruvate. As growth of *S. multivorans* on pyruvate becomes faster upon subculturing, proteome or transcriptome analysis would be an ideal way to get insight into what is changing.

--- At first, the adaptation was a coincidental observation and we did not want to focus too much on it. Nonetheless we think it is a very interesting observation and are thankful for the reviewer to convince us to do additional experiments. We now compared the proteome of adapted cells with that of not adapted cells. The new results are included in the proteome results part. A new supplemental table

showing the proteome results is added. We did not observe a significant change in any of the proteins linked directly to fermentative physiology. However, we found a higher abundance of some proteins also more abundant in the proteomes of *S. multivorans* and *S. cavolei* during fermentative metabolism compared to the respiratory metabolism. Especially the very high abundance and upregulation of a putative aldehyde oxidoreductase under fermentation was observed. We now included a short paragraph on this and three other proteins more abundant in the adapted cells and in fermentative metabolism in the results (L506-521, Table 2) and discussion part (L601 -618) . Whether these proteins are directly involved in fermentative metabolism or contribute to e.g. stress response is subject to further investigations which in our eyes exceed the scope of the current manuscript.

In the manuscript two media are mentioned, one for the pure culture experiments and one for the coculture experiments. The medium for *S. multivorans* (ref 30) is essentially different from the *M. voltae* medium (ref 32). The latter medium contains a large number of organic compounds, which needs to be considered in the experimentation and in the interpretation of the results. By the way, in the publication of Whitman et al different media are presented; it is not clear which of these medium was used in the current study. From the description it is also not clear if both the pure culture and the coculture experiment described in Figure 6 were done in the *M. voltae* medium. It is poorly described how these cocultures were constructed and maintained. What is the inoculum size of each and how is subculturing procedure of the coculture?

--- To avoid any misunderstandings we added a detailed part on the medium composition and cultivation details to the Methods sections:

“The *S. multivorans*/*M. voltae* co-cultures were maintained in modified medium according Whitman *et al.*³². Co-cultures were grown in rubber-stoppered serum bottles with a ratio of aqueous to gas phase of 1:1. The gas phase was N₂/CO₂ [80:20 (v/v); 150 kPa] and lactate (15 mM) the electron donor. Syntrophic co-cultures were initially inoculated in a ratio of *Methanococcus* to *Sulfurospirillum* cells of 5:1. Subculturing was done by transferring 10% (v/v) of the co-culture into freshly prepared media for 10 subcultures before the growth experiment and electron microscopy was performed. The basal medium contained per L 0.1 g isoleucine, 0.1 g leucine, 0.34 g KCl, 4 g MgCl₂ · 6 H₂O, 3.45 g MgSO₄ · 7 H₂O, 0.25 g NH₄Cl, 0.14 CaCl₂ · 2 H₂O, 0.14 g K₂HPO₄, 5 g NaCl. After boiling and cooling to room temperature under N₂, following separately autoclaved anoxic solutions were added (per L basal medium): 10 mL vitamin solution, 10 mL trace element solution (see below), 0.2% (w/v) Casamino Acids (Difco Laboratories, Detroit, Michigan, USA), 10 mL sulfide solution (Na₂S · 9 H₂O, 50 g L⁻¹) and 10 mL cysteine solution (L-cysteine-HCl, 50 g L⁻¹). Final pH was adjusted to 7.0 with 2.5 g L⁻¹ NaHCO₃. The vitamin solution contained per L 2 mg D(+) biotin, 2 mg folic acid, 10 mg pyridoxamine-2HCl, 5 mg thiamine-HCl · 2 H₂O, 5 mg riboflavin, 5 mg nicotinic acid, 5 mg Ca-D(+) pantothenate, 0.1 mg vitamin B₁₂, 5 mg 4-aminobenzoic acid, 5 mg lipoic acid. The trace element solution contained per L 1.5 g nitrioloacetic acid (firstly added, and pH adjustment to 6.5 with KOH), 3 g MgSO₄ · 7 H₂O, 0.5 g MnSO₄ · H₂O, 0.1 g FeSO₄ · 7 H₂O, 0.18 g CoSO₄ · 7 H₂O, 0.1 g CaCl₂ · 2 H₂O, 0.18 g ZnSO₄ · 7 H₂O, 0.01 g CuSO₄ · 5 H₂O, 0.02 g KAl(SO₄)₂ · 12 H₂O, 0.01 g H₃BO₃, 0.01 g Na₂MoO₄ · 2 H₂O, 0.03 g NiCl₂ · 6 H₂O, 0.3 mg Na₂SeO₃ · 5 H₂O, 0.4 mg Na₂WO₄ · 2 H₂O.”

We also added the following sentence to Figure 6: “Pure *S. multivorans* and the co-culture were cultivated in a slightly modified medium originally optimized for *M. voltae* as described in the methods section”

The main organic compounds in this medium are the amino acids leucin and isoleucine as well as casamino acids, we clarified this by adding “A medium optimized for *M. voltae* was used for the co-culture (see Methods section). This medium contained a number of organic substances (i.e. the amino acids leucin and isoleucin, as well as casamino acids) not present in the medium of *S. multivorans* and several controls with *S. multivorans* pure cultures were performed to exclude any unprecedented growth effects potentially caused by this medium.” – see also next point. In the discussion, the following sentence was added: “Since *M. voltae* is not able to thrive in minimal medium³², we cultivated the co-culture not in the medium used for *S. multivorans* cultivation, but in an optimized medium for *M. voltae* which included several organic substances. The influence of this medium on *S. multivorans* cultivation was negligible, so that the observed lactate deprivation and aggregate formation can be unequivocally linked to the syntrophic co-culture.”

What are the growth properties of a pure culture of *S. multivorans* in the *M. voltae* medium? Is the observed aggregate formation caused by syntrophic growth or by the composition of the medium? This needs further description and clarification.

--- Additional experiments showed that growth of *S. multivorans* in the *M. voltae* medium in pure cultures was very similar to that in the “original” medium. We added the sentence “A medium optimized for *M. voltae*³² was used for both co-culture and *S. multivorans* pure culture (see also Methods section). The growth behavior of the latter in this medium with pyruvate and fumarate and pyruvate alone as substrates was similar to that in medium originally used for cultivation of *S. multivorans*.” The aggregate formation is not observed in pure cultures of *S. multivorans* in this medium. We added the sentence and a corresponding figure: “Aggregates were not observed for *S. multivorans* pure cultures in the used media (Supplementary Figure 16).”

The syntrophic behaviour is just discussed as a result of interspecies hydrogen transfer. The option of interspecies formate transfer or direct electron transfer is ignored. This is a shortcoming, especially because *M. voltae* is able to use both hydrogen and formate. The authors mention that the bacterium does not have a pyruvate formate lyase, and that this would make a role of formate less likely. However, formate can also be formed from electrons derived from lactate oxidation and pyruvate oxidation. A proteome or transcriptome analysis of cells of *S. multivorans* grown in pure culture with pyruvate, lactate with fumarate and in coculture with lactate is recommended. Such an experiment would avoid to draw conclusions that are biased. It would also allow to get insight into which enzyme is (enzymes are) involved in lactate oxidation to pyruvate. When lactate would be oxidized by an NAD-dependent enzyme (SMUL_0438), it is also not clear how NADH is oxidized coupled to hydrogen formation. The possible involvement of formate dehydrogenase (Smul-0438) should be considered.

--- Since we never detected formate in any of the HPLC measurements of any culture (and concluding from standard measurements with formate we should have been able to detect it), we did not consider the possibility of interspecies formate transfer and still think that it is unlikely. This holds especially true, since the fermentation balance is completely even in both *Sulfurospirillum* spp. To make more clear we didn't detect formate which is therefore obviously not a metabolite produced during fermentation we now wrote "No other organic acids such as formate, or alcohols e.g. ethanol, were detected". We would include direct electron transfer as a possibility if we would have found either 1) structures for DIET in the electron micrographs or 2) one of the archetypical proteins (e.g. the large Geobacter-like outer surface cytochromes or any pili) encoded in one of the *Sulfurospirillum* spp. genomes. Both were not the case, so we did not include it in the manuscript. We added a discussion on both: "The interspecies electron carrier in the syntrophic relationship of *S. multivorans* and *M. voltae* is hydrogen based on the metabolite analysis of pure *S. multivorans* cultures, where only hydrogen was found as a product of pyruvate fermentation. Direct interspecies electron transfer is also unlikely to proceed in this syntrophic interaction, since besides none of the typical nanowire structures were visible in electron microscopy, no pili or extracellular cytochromes⁶⁰, usually used for interspecies electron transfer of e.g. *Shewanella* spp. and *Geobacter* spp.⁶¹ were found in the genome of any *Sulfurospirillum* spp."

We could also not measure any NAD(P)(H)-dependent lactate oxidation or production, which is also stated in the manuscript, we added the complete methods in the supplement (Line 476 "...lack of pyridine dinucleotide-dependent lactate-oxidizing or pyruvate-reducing activities in cell extracts of *S. multivorans* (data not shown, methods described in Supplementary Note 2)"). There is also evidence in the literature that such an enzyme is not involved in lactate metabolism of the closely related *Campylobacter jejuni* (see reference 50, "Two respiratory enzyme systems in *Campylobacter jejuni* NCTC 11168 contribute to growth on L-lactate").

However, the reviewer is right in that proteomics could give helpful insights into the involvement of lactate oxidating enzymes. Therefore we investigated the proteome of lactate-cultivated cells in comparison to the proteome of cells grown with pyruvate, both with fumarate as electron acceptor. The results are now added as supplementary excel file and added to the results text (L.484-494). However, we could not find an upregulation of the putative lactate-oxidizing enzymes, therefore we suggest a substrate-independent expression of those, which is in line with the observation of the corresponding enzymes in *C. jejuni* (Ref 50). We included a brief discussion as follows:

"A role of the *S. multivorans* "lactate dehydrogenase" in respiratory lactate oxidation was also unlikely as its low abundance in the proteome of lactate-grown cells suggest. Instead, lactate is likely oxidized in *S. multivorans* by orthologs of the NAD⁺-independent enzymes recruited by *C. jejuni* for lactate oxidation, a flavin Fe-S cluster-containing enzyme and a three-partite lactate utilization protein⁵⁰. These proteins were shown to be not substrate-inducible in *C. jejuni*, which is in line with the observed similar abundance of both proteins in lactate and pyruvate-cultivated *S. multivorans* cells."

Hydrogenase activity measurements were done with methylviologen and benzylviologen. Is there any

activity with NADH or NAD in the hydrogen formation and hydrogen oxidation reaction, respectively. Was it possible to detect formate dehydrogenase activity in cell extracts?

--- We did not detect any hydrogenase activity with NADH, which is corroborated by the lack of any typical NAD(H) dependent hydrogenase in the genome of *S. multivorans*. (see also Kruse et al., 2017, Front Microbiol, where we tested H₂-oxidizing activities). For sure there is formate dehydrogenase activity in the cells, since it is expressed under all conditions tested yet (Goris et al. 2015 Sci Rep, this manuscript) and an Fdh was characterized before (Schmitz & Diekert, 2003, Arc Microbiol). However including this in the manuscript does in our eyes not contribute to the discussion, since we have not detected formate under any physiological condition – even more so given the highly artificial conditions of Fdh activity measurements.

Minor point:

In the introduction it is mentioned that hydrogen production was never shown for Epsilonbacteria. This statement is not completely true. Hydrogen production was shown for *Sulfurospirillum carboxydovorans* (Jensen and Finster (2005) *Antonie van Leeuwenhoek* 87: 339–353. This is reference 28 of the manuscript.

--- True, we overlooked this here since we mainly searched for fermentative hydrogen production. We added now “*Sulfurospirillum carboxydovorans* was shown to produce minor amounts of hydrogen, which was finally consumed again, upon CO oxidation²⁸.” Also, we deleted “exclusively” in front of H₂ oxidizing bacteria and added “fermentative” to “hydrogen production was never observed”.

Reviewer #2 (Remarks to the Author):

The authors show that some *Sulfurospirillum* spp., which had previously been viewed only as hydrogen oxidizers, produce hydrogen during pyruvate fermentation. In addition, *S. multivorans* was shown to grow syntrophically on lactate when a hydrogenotrophic methanogen, *Methanococcus voltae*, was present. Enzyme assays reveal that hydrogen-producing activity was present in the membrane fraction and likely cytoplasmically oriented. Hydrogen production activity was correlated to hydrogenase 4 (Hyf) based on proteomic and transcriptional analyses. Hyf gene cluster was detected in genomes of other *Sulfurospirillum* and in some *Campylobacter* species.

Interestingly, it took many transfers for *Sulfurospirillum* spp. to ferment pyruvate and produce hydrogen effectively. Some *Sulfurospirillum* spp. produce mainly acetate, hydrogen, and carbon dioxide from pyruvate while other produce less hydrogen due to their ability to produce fermentation products such as succinate. The work on fermentation products was done well but the results are not surprising.

Specific comments:

1. The authors concluded that some members of the Epsilonproteobacteria should be considered to be hydrogen producers in their natural habitats. It is likely that some will grow syntrophically on lactate as shown for *S. multivorans*. However, the ecological importance of hydrogen production from pyruvate is not clear. It took many transfers in the laboratory before *Sulfurospirillum* spp. effectively fermented pyruvate and produced hydrogen. This argues that the ecological role of these organisms is not pyruvate fermentation. In general, *Sulfurospirillum* spp. do not oxidize or ferment carbohydrates so fermentative hydrogen production from pyruvate derived from sugars is not likely.

--- We agree with the author in that the amount of pyruvate in natural habitats is most likely rather low and thus its ecological role as substrate is limited (despite cell lysis could provide some pyruvate also in natural habitats). Therefore we also follow the reviewer that lactate is the main substrate in natural habitats. This we underline in our manuscript by setting up the co-cultures of *S. multivorans* with *M. voltae* with lactate instead of pyruvate, which was not possible for pure *S. multivorans* cultures. This we underline in the manuscript by adding "The inability of *Sulfurospirillum* spp. to use lactate as sole substrate in pure cultures is most probably due to the thermodynamically unfavorable lactate oxidation to pyruvate upon H₂ production. To test the ecological significance of our observation, we established a lactate-consuming syntrophic partnership of *S. multivorans* with a hydrogen-consumer, *Methanococcus voltae*." However, to reveal the details of the fermentative physiology of *Sulfurospirillum* spp., we looked at pyruvate fermentation.

One reason for the relatively long adaptation process could be that the bacteria are cultivated usually for a very long time in a respiratory mode, in our laboratory as well as in cell culture collections. We added the following sentence to discuss this: "The low adaptation to fermentative metabolism in *Sulfurospirillum* might also be an effect of the long-term respiratory cultivation of the organisms in our laboratory and cell culture collections. Whether *Sulfurospirillum* spp. in natural habitats show an equally long adaptation time to fermentative metabolism is likely dependent on the type and concentration of electron donor and electron acceptor in the environment."

Indeed the low amount of hydrogen produced by not fermentatively adapted *S. multivorans* cells could hinder the syntrophic relationship. Therefore we performed additional experiments with *S. multivorans* cells not adapted to pyruvate fermentation and cultivated them with *Methanococcus voltae*. To our surprise, the growth/lactate consumption was not much slower (see Supplementary Figure 16) than with the adapted cells, which points towards a higher ecological relevance of fermentation of *Sulfurospirillum*. Therefore we added a sentence to the discussion in L 691, "Both, *S. multivorans* cells adapted and not adapted to pyruvate fermentation supported growth of the co-culture, which strengthens our suggested role of *Sulfurospirillum* spp. as H₂ producers in anaerobic food webs."

However, of course we agree with the reviewer in that fermentation is only one of many lifestyles of *Sulfurospirillum* and other free-living Epsilonproteobacteria. Therefore we added an addendum to the conclusion: "All these findings imply an even higher versatility for Epsilonproteobacteria than previously thought, also it should be noted here, that most of them are primarily rely on a respiratory metabolism."

2. The authors provided proteomic data in supplemental tables excel files, but these data should also be deposited in a database such as PRIDE.

---We deposited the proteomic raw data in PRIDE (Accession number PXD010316, **Username:** reviewer73785@ebi.ac.uk **Password:** IFd6Gnmr)

Minor comments:

1. p. 33. Spell out genus name here.

--- Done

2. Line 109. I would state that the gas volume was adjusted to standard temperature and pressure.

--- We added "The gas volume was adjusted to standard temperature (25° C) and pressure (1 bar)"

3. Line 110. Was medium used for the adaptation experiment the same as that described on lines 94-95?

--- Yes. We made that clear by adding "in the *S. multivorans* medium as stated above"

4. Line 181. manufacturer's

--- corrected

5. Line 231. There is no period at the end of the sentence making me wonder if there is some text missing.

--- Period was added.

6. Lines 244-246. This statement seems out of place as the data are discussed in the next section.

--- Here, we deleted the second part from L 245. We keep L 244 f, but deleted the corresponding statement in the next section.

7. Lines 259-261. Cite Fig. 2

--- Cited

8. Line 330-331, periplasmically oriented and cytoplasmically oriented would be better wording, as both are located in the cytoplasmic membrane.

--- Thanks for the suggestion, changed accordingly.

9. Line 350-445. Fig. 5 needs to be cited. The order of topics discussed could be improved. I suggest discussion of the data on hydrogenases together with the data on Hyf first (lines 385-403) to make it clear that Hyf is the one involved in hydrogen production. One can then discuss pyruvate/energy metabolism and anabolism.

--- We changed the structure accordingly and made a few changes to enhance the flow. Fig 5 is cited appropriately in the Lines 397, 413, 436, 439, 463 of the revised manuscript

10. Lines 400-403. The authors state that an analysis of the amino acid involved in proton translocation was done but they do not state what they conclude from these data.

--- We added a brief conclusive sentence here, since we already wrote about that topic in the discussion part. "Important conserved amino acids which are likely involved in proton-pumping of complex I are conserved in one membrane subunit of Hyf, namely in HyfF"

Michael J. McInerney

Reviewers' comments:

Reviewer #1 (Remarks to the Author):

I have read the revised version of the manuscript and the reply to the comments of the reviewers. I am satisfied with the changes that were made.

My point concerning the possibility of formate transfer was not yet taken away. The authors mention that they did not detect formate in the cultures. However, in syntrophic cultures the levels of hydrogen and formate will be low. Low hydrogen levels can be measured, but low formate levels not. What was the detection limit of formate in the analysis? Based on measured hydrogen levels in cultures measured and assuming equilibrium of $H_2 + HCO_3^- \rightarrow Formate^- + H_2O$ the authors can calculate what the formate concentration would be and compare that with the detection limit of their analysis. I recommend to make these calculations to avoid biased conclusions.

Reviewer #2 (Remarks to the Author):

I find that the points raised in the previous round of review have been satisfactorily addressed by the authors. They conducted additional experiments on pyruvate-adapted and not adapted cells to determine the potential proteins involved in the adaptation process. They also performed experiments with non-adapted cells to ensure that these cells would also form syntrophic associations with methanogens. The text has been modified so that their conclusions are more conservatively stated. I have a few editorial corrections.

1. Line 60. The statement on DIET lacks does not have reference(s), which should be added.
2. Lines 550-554. This is a very long sentence. Does all of the statements refer to only SNUL_2101 or are there several proteins being discussed here?
3. Line 580 (see also Methods section on media). The German rather than English spelling of leucine and isoleucine are used.
4. Line 693. ...dehydrogenase was not detected or was in low abundance in the proteomes....
5. Line 728. Degradation rather than deprivation
6. Line 758. Some words are missing here after chemiosmoto...

Michael J. McNerney

Reviewers' comments:

Reviewer #1 (Remarks to the Author):

I have read the revised version of the manuscript and the reply to the comments of the reviewers. I am satisfied with the changes that were made.

My point concerning the possibility of formate transfer was not yet taken away. The authors mention that they did not detect formate in the cultures. However, in syntrophic cultures the levels of hydrogen and formate will be low. Low hydrogen levels can be measured, but low formate levels not. What was the detection limit of formate in the analysis? Based on measured hydrogen levels in cultures measured and assuming equilibrium of $H_2 + HCO_3^- \rightarrow Formate^- + H_2O$ the authors can calculate what the formate concentration would be and compare that with the detection limit of their analysis. I recommend to make these calculations to avoid biased conclusions.

The reviewer is right, in our set-up we cannot exclude that low levels of formate are formed by *Sulfurospirillum*. Our detection limit with the used HPLC column and settings was about 0.5 mM, which is much higher than the concentration of about 5 μ M which we calculated for the co-culture assuming of $H_2 + HCO_3^- \rightarrow Formate^- + H_2O$ with +3kJ/mol. The production of formate is still rather unlikely in our view, since no formate dehydrogenases similar to the described formate-producing ones (or of course a PFL) were found in any *Sulfurospirillum* genome. Therefore we added a careful and brief discussion on that topic to the manuscript (L690 to 697): "However, lower levels of formate (< 0.5 mM) could be not detected in our analysis and therefore we cannot completely exclude formate as a second electron carrier of lower importance here. Production of formate from CO_2 in *S. multivorans* could be mediated by a cytoplasmic formate dehydrogenase (SMUL_0079), for which the electron donor and catalytic bias is yet unknown. The three periplasmic formate dehydrogenases of *S. multivorans* are putatively formate-oxidizing and most probably connected to the quinone pool via a cytochrome *b* subunit and therefore the production of formate by these enzymes would be thermodynamically unfavorable, since quinones have a relatively positive midpoint potential of around -100 mV."

Reviewer #2 (Remarks to the Author):

I find that the points raised in the previous round of review have been satisfactorily addressed by the authors. They conducted additional experiments on pyruvate-adapted and not adapted cells to determine the potential proteins involved in the adaptation process. They also performed experiments with non-adapted cells to ensure that these cells would also form syntrophic associations with methanogens. The text has been modified so that their conclusions are more conservatively stated. I have a few editorial corrections.

1. Line 60. The statement on DIET lacks does not have reference(s), which should be added.

- Reference now included.

2. Lines 550-554. This is a very long sentence. Does all of the statements refer to only SMUL_2101 or are there several proteins being discussed here?

- This sentence is now divided in two sentences, also to make clear that only the protein encoded by SMUL_2101 is discussed here.

3. Line 580 (see also Methods section on media). The German rather than English spelling of leucine and isoleucine are used.

- Now corrected.

4. Line 693. ..dehydrogenase was not detected or was in low abundance in the proteomes...

- This suggestion is now inserted

5. Line 728. Degradation rather than deprivation

- Changed

6. Line 758. Some words are missing here after chemiosmoto...

- Now added: chemiosmotic coupling.

Michael J. McInerney